# Vasa nucleates asymmetric translation along the mitotic spindle during unequal cell divisions

Ana Fernandez-Nicolas[1,2], Alicia Uchida[1,2], Jessica Poon[1] & Mamiko Yajima [1✉]

mRNA translation on the spindle is hypothesized to be an essential strategy for the localized production of cell regulators. This mechanism may be important particularly in early embryonic cells, which have a large diffusion volume and that undergo rapid cell divisions. Evidence to test such a hypothesis has been, however, limited. Here, we use an embryo with both symmetric and asymmetric cell divisions and manipulate Vasa protein, an RNA-helicase, on the spindle in live sea urchin embryos. We learned that the spindle serves as a major site of translation and that protein synthesis within a single spindle can be unequal and help drive asymmetric cell divisions during embryogenesis. Recruiting Vasa to the ectopic sub-cellular region induced a new site of translation, disturbed asymmetric translation on the spindle, and changed the cell fate. Based on these observations, we conclude that Vasa functions in localized translation, which provides a spatiotemporal control in protein synthesis and is essential for rapidly developing embryonic cells.

---

[1] Department of Molecular Biology Cell Biology Biochemistry, Brown University, 185 Meeting Street, BOX-GL277, Providence, RI 02912, USA. [2] These authors contributed equally: Ana Fernandez-Nicolas, Alicia Uchida. ✉email: mamiko_yajima@brown.edu

Differential protein distribution in the cell enables functional specializations required for complex processes of homeostasis. During development though, when cells are transitioning from various cell states, differential protein distribution has a significant impact on intercellular signaling, cell fate specification, and morphogenesis[1,2]. A common mechanism for protein localization in developmental processes includes direct targeting/modifications of the protein via specific signal sequences (e.g., membrane-targeting or nuclear localization signal), yet not all protein localizations are autonomously determined. In particular, proteins functional in the cytoplasm usually have no distinct localization sequence, yet they often accumulate in specific regions of the cytoplasm that enables efficient functionality. This may result from the translation of localized mRNAs, or a localized translation of general mRNAs. The mitotic spindle is a subcellular region where specific mRNAs are reported to localize[3–5]. Studies in *Xenopus*, sea urchins and mouse oocytes have shown that ribosomes and active translation machinery are tightly associated with mitotic microtubules[5–11], implying wide conservation of spindle-associated translation, yet neither its biological significance nor molecular mechanism is clear at this time.

In this report, as a promising factor that regulates localized translation on the spindle, we focus on Vasa, a DEAD-box RNA helicase that is conserved among all metazoans[12–14]. Vasa is implicated in translational regulation[15–17] and involved in mitotic regulation[18,19]. It is enriched on the mitotic spindle of every blastomere of the sea urchin embryo until the 8-cell stage and is essential for cell cycle progression during embryogenesis as well

as in the germline later in development (Fig. 1a; Movie S1)[19,21]. Vasa knockdown in those embryos results in cell division defects and ~80% reduction of general translation in the embryo, implying its function as a general translation regulator[22]. At the 8–16 cell stage, the first asymmetric cell division results in large and small cells (macromeres and micromeres, respectively) that have distinct cell fates. Vasa protein is associated with this asymmetry, transitioning from its uniform distribution along the mitotic spindle to become enriched toward the future micromere-domain. This asymmetry is maintained throughout the remainder of the division, resulting in enrichment of Vasa protein specifically in micromeres at the 16-cell stage and eventually into small micromeres, the small cells of the next asymmetric division, to form the germline of this animal[19]. This repeated asymmetric regulation of Vasa yields a Vasa-null cell (large micromere) with a singular fate of skeletogenesis, and a Vasa-positive cell (small micromere) with a totipotent, germ cell fate, all within 40 min for each cell cycle. Intriguingly, *vasa* mRNA is also enriched on the spindle during the cell division[22] (Supplementary Fig. S1a), suggesting that a population of mRNAs may be recruited for on-site translation on the spindle. This tilting of a previous balance of translation along the spindle may contribute to the asymmetric molecular distribution of newly synthesized proteins. In this study, we report that Vasa is responsible for facilitating localized translation at a specific sub-cellular region by recruiting mRNAs and translation machinery, which is essential for proper cell divisions and fate determinations during embryogenesis of the sea urchin.

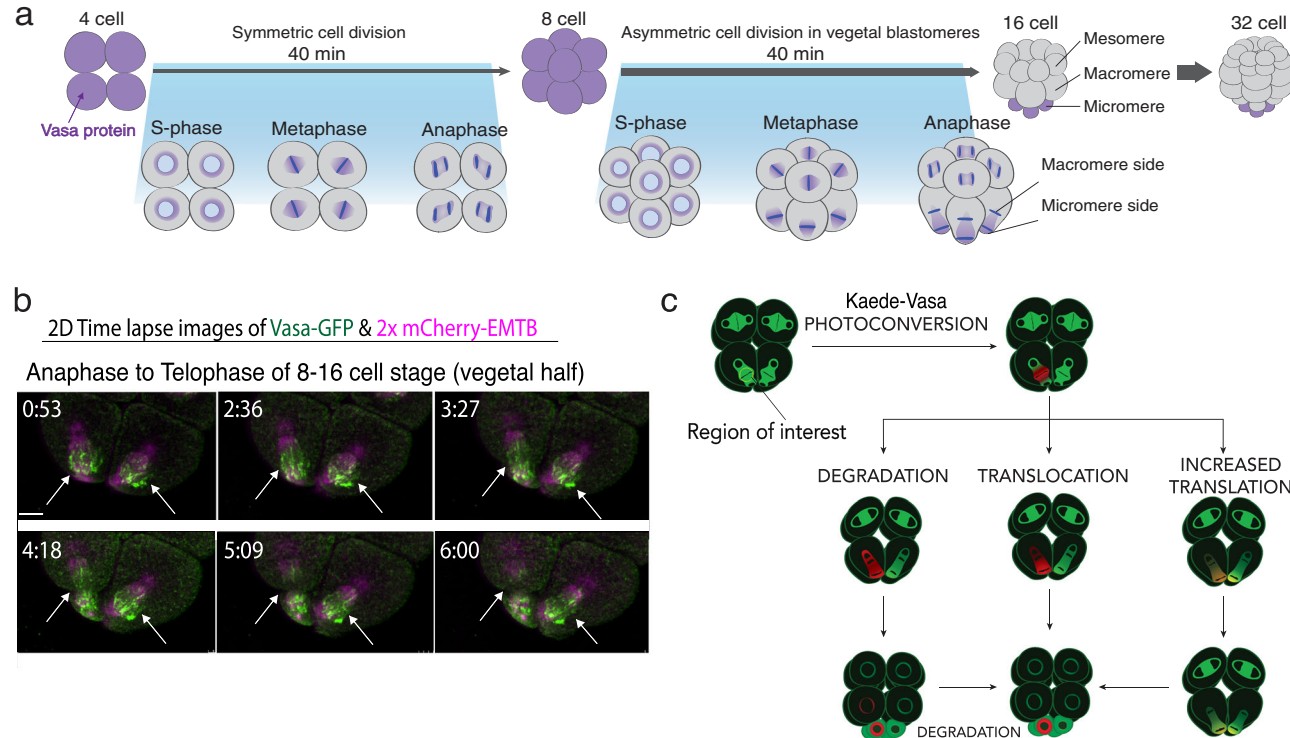

**Fig. 1 Vasa dynamics during asymmetric cell division. a** Vasa distribution in early embryogenesis. **b** Time-lapse images of the vegetal half of the Vasa-GFP (green) embryo during micromere formation (arrows) at the 8–16-cell stage presented as maximum 2D-projection. 2xmCherry-EMTB (magenta)[20] was used as counter-staining to visualize microtubules. These experiments were performed at least three independent times. Scale bars = 10 μm. **c** Kaede-Vasa enrichment model. Unphotoconverted Kaede-Vasa (green) on the spindle around the chromosomes is photoconverted and emits red fluorescence. Dynamics of photoconverted Kaede-Vasa were monitored over time to determine if Vasa enrichment in the micromeres was accomplished through differential degradation, translocation, or upregulation of translation on the micromere side of the spindle. If decreasing levels of photoconverted Kaede-Vasa on both sides of the spindle is complemented by an increase in unphotoconverted Kaede-Vasa on the micromere side of the spindle during anaphase, it suggests that upregulated translation is responsible for the asymmetric enrichment of Vasa in micromeres.

## Results and discussion

**Asymmetric Vasa protein synthesis occurs along the spindle.** During asymmetric cell division at 8–16 cell stage (micromere formation), Vasa protein first localizes symmetrically on both sides of the spindle at metaphase yet becomes more enriched toward the micromere-side of the spindle during anaphase and telophase (Fig. 1b)[19]. To gain the quantitative accuracy, embryos injected with Vasa-GFP were imaged live during metaphase of the 8–16 cell stage by Resonance fast scanning microscopy, which requires minimum laser power and scanning time and turned out to be less toxic even compared to the multi-photon microscopy[23], with a complete z-stack of 0.44um-interval. This setting gave enough resolution for 3D-reconstruction of the stacked images without notable cytotoxicity such as cell cycle delay or cell division abnormality (Supplementary Fig. S1b; Movie S2).

To determine how Vasa protein becomes asymmetric on the spindle, Kaede-Vasa mRNA was used. Kaede that naturally fluoresces green irreversibly turns in red by a UV light[24,25], making Kaede a useful tool to distinguish pre-existing protein (red) from newly translated protein (green) [26,27]. We previously demonstrated that Kaede-Vasa mimics endogenous Vasa protein localization and its photoconversion is sensitive enough to monitor the protein movement at the sub-cellular level without causing any apparent cellular toxicity or bleaching in the developing sea urchin embryo[23]. Further, Kaede-Vasa appears to successfully mimic the endogenous function of Vasa in cell cycle progression (Supplementary Fig. S2a). To be noted, this Vasa localization on the spindle is distinct from the non-specific signal enrichment in the nucleus occasionally seen with the fluorescent dye (Supplementary Fig. S2b) [28].

At least three different mechanisms may contribute to local protein enrichment; (1) protein degradation outside of the specific region, (2) translocation of the protein to the site, and (3) localized translation in a specific region of the cell (Fig. 1c). To identify which mechanisms are causative for Vasa protein asymmetry on the spindle, Kaede-Vasa on the spindle of either animal or vegetal blastomere was photoconverted and imaged during metaphase to anaphase of the 8–16 cell embryo. Under this experimental condition, the specific photoconversion only on the spindle was confirmed by the 3D rotation of the image immediately after photoconversion (Supplementary Fig. S2c, arrows; please see also the methods and Movie S3 for details and rotation images, respectively).

In animal blastomeres that undergo symmetric cell division (Fig. 2a, b), the photoconverted region was initially intensely red (Fig. 2a, red circle) but decreased rapidly while the green intensity level recovered over time symmetrically on the spindle, suggesting pre-existing Vasa (red) decreased but newly synthesized Vasa increased on the spindle overtime during M-phase. Slopes for the trend lines of the resulting graphs showed that the average side 1/ side 2 ratios of photoconverted Kaede-Vasa (red) in the animal (symmetric) blastomeres were 1 (Fig. 2c). This result suggests that the Vasa clearance rate was approximately equal on both sides of the spindle during symmetric cell division. In vegetal blastomeres that undergo asymmetric cell division (Fig. 2d, e), unexpectedly, the Vasa clearance rate was also equal on each side of the spindle. The average micromere/ macromere ratio of the photoconverted Kaede-Vasa intensity level was 1.06 during anaphase, similar to that found in the animal blastomeres (Fig. 2f; Movie S4). These results suggest that Kaede-Vasa was cleared at a similar rate on the spindle of every blastomere. Therefore, the difference in Vasa accumulation during the asymmetric cell division may not be the result of differential Vasa degradation. On the other hand, the newly synthesized (green) Kaede-Vasa signals were found to increase only on the micromere-side of the spindle during Anaphase (Compare A1/A2 and V/A ratios in the column graphs

in Fig. 2c, f) throughout three independent experiments. Based on these observations, we speculate that upregulation of newly synthesized Vasa on the micromere-side is caused either by translocation of Kaede-Vasa from elsewhere or by increased protein synthesis of Kaede-Vasa specifically on the micromere-side of the spindle during asymmetric cell division.

To test if any translocation of Vasa protein contributes to asymmetric Vasa localization during micromere formation, we photoconverted only one side of Kaede-Vasa on the spindle both during symmetric and asymmetric cell divisions (Fig. 3a, b; Movie S5). We detected no significant translocation of converted Kaede-Vasa from one side to the other side of the spindle during the M-phase (Average slope ratio of side 1 and side 2 at 0.995 or 1.019 for asymmetric or symmetric cell division, respectively; Fig. 3c). To identify a possible translocation between cytoplasm and spindle, a portion of the cytoplasm was also photoconverted during M-phase (Supplementary Fig. S3a–e; Movie S6) or S-phase (Fig. 3f–j; Movie S7). As a result, during M-phase, no active translocation of the Kaede-Vasa signal to the spindle was observed, yet a notable signal increase was observed at a time of M-phase entry. Similarly, when Kaede-Vasa on the spindle was photoconverted and tracked over an entire cell cycle division (Supplementary Fig. S4a–e), the Kaede-Vasa signal in the cytoplasm increased slightly during S-phase and sharply increased at a time of M-phase entry on the newly formed spindle. Further, to track the dynamics of cytoplasmic Kaede-Vasa, multiple photoconversions (~8 repetitive photoconversions) were performed to photoconvert the majority of the cytoplasmic Kaede-Vasa in the cell during asymmetric cell division (Fig. 3d–f and Supplementary Fig. S4f, g). The photoconverted Kaede-Vasa level in the spindle area remained low during M-phase yet sharply increased at S-phase entry, suggesting that cytoplasmic Vasa may be also sustained in the cytoplasmic structures during M-phase.

Taken together, it appears that active translation of Kaede-Vasa occurs during M-phase on the spindle and is asymmetrically increased on the micromere-side of the spindle during asymmetric cell division. Vasa both on the spindle and in the cytoplasm showed little mobility during M-phase yet were released during S-phase in the cell, which then at least partly translocate back to the spindle at the time of the next M-phase entry (Fig. 3g). Further, no significant Vasa protein translocation was observed during M-phase between the cytoplasm and the spindle or even within a single spindle. Therefore, we propose that the increased Vasa signal on the micromere-side of the spindle was regulated by differential on-site translation of Kaede-Vasa mRNA. The spindle may serve as a major site of translation at least during M-phase and this selective translation site can also reveal molecular asymmetries.

**The general translation is enriched on the spindle during mitosis and results in asymmetric protein accumulation.** To test if general mRNA translation is also increased on the micromere-side of the spindle, we visualized general translation activities in the cell using homopropargyl-glycine (HPG) or O-propargyl-puromycin (OPP) as in vivo reporters of protein synthesis[29,30]. The spindle area was identified by Vasa using its nature of enrichment on the spindle. Embryos were treated with HPG or OPP during the 8–16 cell stage transition for 15 ~ 30 min prior to fixation (Supplementary Fig. S5a–c). As the OPP itself stops translation once bound to the translation machinery in the cell, the titrated level of OPP was added once the embryo entered the M-phase not to halt a cell cycling. OPP intensity on the spindle region (counterstained by the Vasa signal) compared to cytoplasmic regions was ~2.12-fold higher (Fig. 4a, b; ~68% of the total protein synthesis is on the spindle region), and HPG

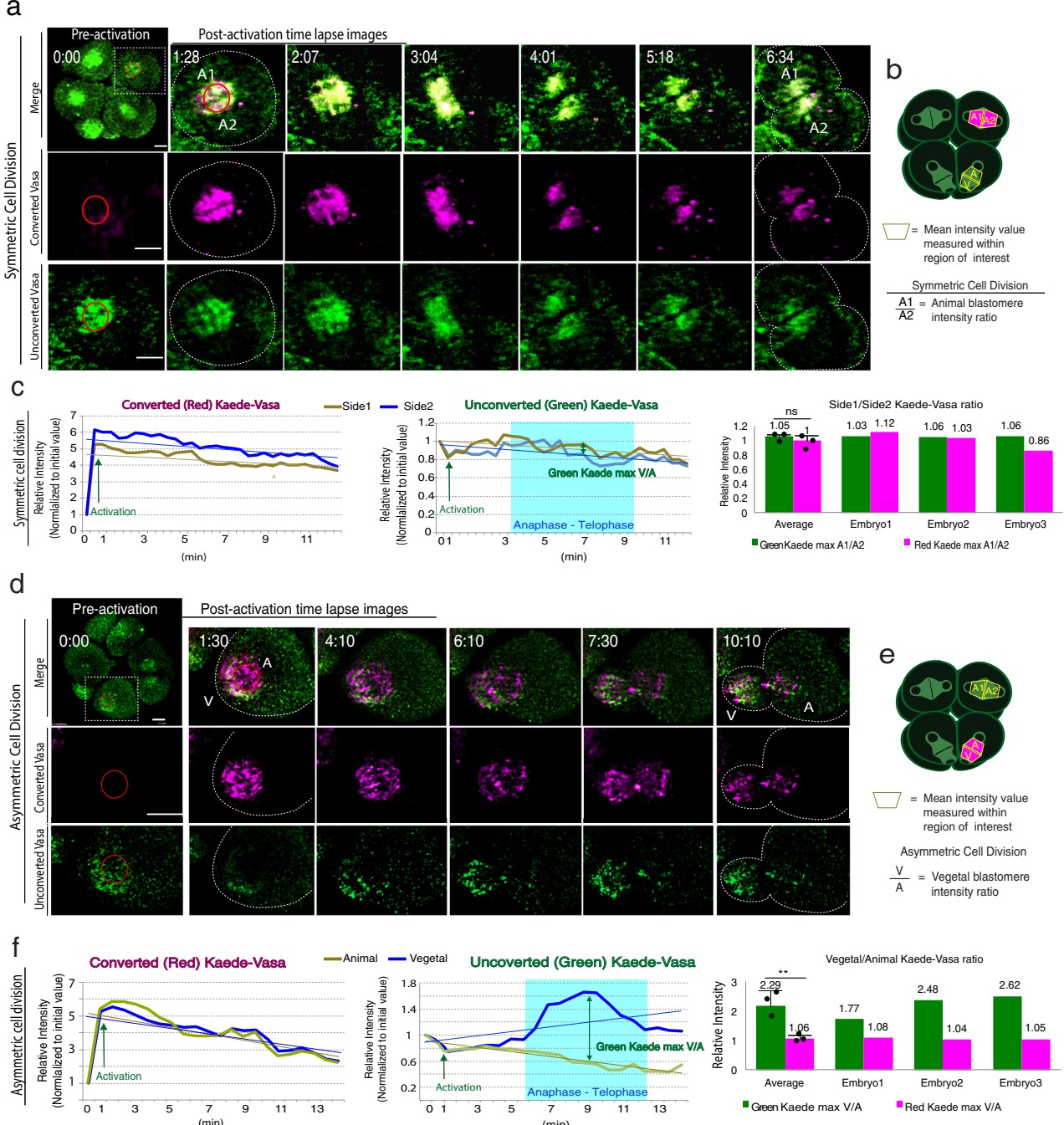

**Fig. 2 Photoconverted and unphotoconverted Kaede-Vasa dynamics during symmetric (a) or asymmetric (d) cell division. a, d** A whole Z-section of the cell (0.44 µm per slice; ~90 slices in total) was imaged and analyzed every 1-min and presented as maximum 2D-projection. The upper leftmost column shows Kaede-Vasa expression prior to photoconversion (green) while the remaining columns are magnified images of the region squared by a white dashed line. Images were taken at varying minute intervals as indicated in the corner, following photoconversion (magenta) during cell division. The area marked by a red circle is a photoconverted area (activation ROI). Kaede-Vasa levels were compared between the vegetal side (V) and the animal side (A) of the spindle during asymmetric cell division (B) or both sides of animal blastomeres (A1 and A2) during symmetric cell division (A). Scale bars = 10 µm. **b, e** A cartoon diagram depicting a measurement area and protocol for comparing fluorescent signal intensity across the mitotic spindles in either the animal (**b**) or vegetal blastomeres (**e**). Photoconversion was performed at an 8-cell stage metaphase. The major photoconverted area (magenta) typically appears in a trapezoid by following the shape of the spindle, which is labeled as A, V, A1, or A2 and indicates the area of the intensity measured (Detection ROI), respectively. The relative intensity of A1 over A2 for symmetric cell division and V over A for asymmetric cell division was compared and shown in graphs (**c**) and (**f**). **c, f** Kaede-Vasa intensity level during symmetric (**c**) and asymmetric (**f**) cell division was measured for detection ROI and normalized to its initial value. The green Kaede-Vasa was maximum ~2.29 folds higher in the vegetal side (V) compared to the animal side (A) of the spindle during asymmetric cell division, whereas nearly equal in both sides of the spindles (A1 and A2) during symmetric cell division (*p*-value, 0.0094). All experiments were performed at least three independent times and the representative analyses are shown (*n* = 3 each). Unpaired *t*-test was used for all statistical analyses in this figure. **\*\****p* < 0.01. Columns represent means ± SD or SEM. Source data are provided as a Source Data file.

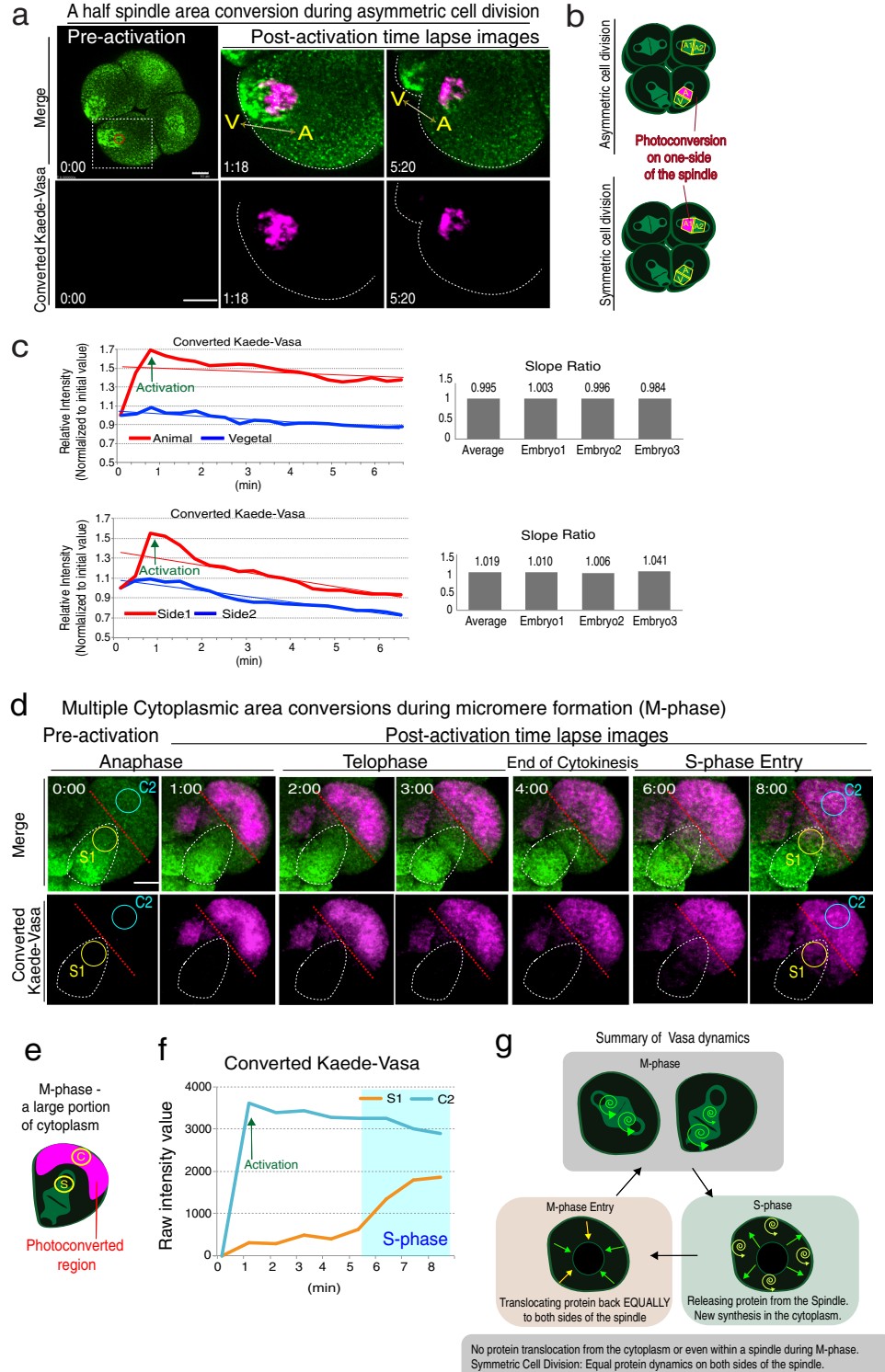

incorporation on the spindle was ~1.85-fold higher (Fig. 4d, e; ~59% of the total protein is on the spindle region) than in the cytoplasm during the M-phase. Taking into account the different areas occupied by the spindle region versus the cytoplasm, and the background level measured in the presence of emetine, a translation blocker, the spindle area generates ~1.45-fold (~46.5%) more protein synthesis than random areas of the cytoplasm.

During the 8–16 cell transition, we found that the ratio of both OPP and HPG levels was nearly equal on both sides of an animal blastomere (symmetric) spindle region throughout the M-phase of the cell cycle (Fig. 4a, d, arrows). In vegetal (asymmetric) blastomeres, on the other hand, HPG and OPP levels were higher on the micromere side than on the macromere side from early anaphase to telophase (Fig. 4c, f). Importantly, this asymmetry was even more pronounced for Vasa protein distribution, so that the fluorescence intensity levels of Vasa in the micromeres was approximately 2.9 times higher than that of the macromeres at Anaphase (Fig. 4e). To be noted, the asymmetry of the OPP and HPG signals was much reduced and the signal localization was in

**Fig. 3 Kaede-Vasa does not translocate within a spindle or from the cytoplasm. a** Kaede-Vasa was photoconverted on one side of the spindle at the 8-cell stage metaphase during asymmetric cell division (A or V) or during symmetric cell division (A1 or A2). These experiments were performed at least three independent times and a representative image series of the embryo is shown. The cell was photoconverted only for the animal side (A) of the spindle during asymmetric cell division. Scale bars = 10 μm. **b** Diagrams indicating where photoconversion was performed (magenta) and detection ROIs were measured. **c** Photoconverted Kaede-Vasa (red line) showed a significant increase immediately after photoconversion yet decreased at a similar rate with that on the other side of the spindle, suggesting Kaede-Vasa did not translocate to the other side of the spindle yet rather remained at the original site during M-phase. Each experiment was repeated at least three times and a representative slope ratio out of three individual embryos is shown in column graphs on the right. Each slope ratio was nearly equal on both sides of the spindle. **d** A majority of Kaede-Vasa in the cytoplasmic region was photoconverted during M-phase. A time series of the vegetal blastomere images undergoing asymmetric cell division is shown. **e** A diagram indicating where photoconversion was performed (magenta) and detection ROIs (S1, C2) were measured. Scale bars = 10 μm. **f** Photoconverted Kaede-Vasa remained in the cytoplasm (C2) during M-phase, yet the signal was increased in the spindle region (S1) after S-phase entry. These experiments were performed at least three independent times and the measurement results of two other embryos are shown in Supplementary Fig. S4f, g. **g** A summary model of Vasa dynamics during cell cycle progression. Increased Vasa protein synthesis occurs on the vegetal side of the spindle during asymmetric cell division. Vasa is released from the sub-cellular structures (e.g., microtubules) upon S-phase entry and partly translocates back to the spindle at the beginning of the next M-phase. Source data are provided as a Source Data file.

general broader compared to that of Vasa. This is likely due to a relatively long incubation time (15 ~ 30 min that covers most of the M-phase) necessary for signal visualization of OPP or HPG. However, we also could not exclude a possibility some translation may occur outside of the spindle area. A similar trend was also seen in the localization of Phospho S6 Ribosomal protein that detects active translation machinery (Supplementary Fig. S5d–g)[13]. These results suggest that general translation, including Vasa translation, was increased on the micromere-side of the spindle region during asymmetric cell division and is consistent with the results of photoconversion experiments (Figs. 2–3). These results demonstrate increased active translation complexes on the micromere-side of the spindle, which might contribute to the increased translation of Vasa (and other proteins) on site. Micromeres are known to enrich various molecules and function as a major organizer at the 16-cell stage[31,32]. Since Vasa functions as a general translation regulator in this embryo[19,22], its asymmetry may contribute to differential cell fate during asymmetric cell division.

**Vasa mRNA translation is initiated on the spindle**. The above observations suggest that Vasa may be asymmetrically translated on the spindle during micromere formation, contributing to differential cell fate specification. To directly test when and where Vasa translation is initiated in real-time, in vivo tetracysteine detection system was used (Fig. 4g). In this approach, the sequence encoding the tetracysteine (TC) tag is inserted immediately downstream of the start site for translation of the mRNA of interest. Upon translation activation of the tagged mRNA, a biarsenical dye preferentially binds to the nascent TC tag associated with ribosomes and immediately fluoresces (*red* with ReAsH and *green* with FlAsH reagents), allowing real-time detection of translation activities of the tagged mRNAs in live cells (Supplementary Fig. S6a, b)[33–35]. Upon ReAsH introduction at the 8–16 cell stage, TC-Vasa-GFP but not Vasa-GFP (negative control) emitted red fluorescence within 3~6 min of introduction on the spindle, consistent with the endogenous Vasa location (Fig. 4h and Supplementary Fig. S6c, d). The TC-ReAsH signal increased over time and remained on the spindle during recording and was higher on the micromere side of the spindle. These results independently support the contentions that 1) translation indeed occurs on the spindle, 2) translation is higher on the micromere-side of the spindle, and 3) translated Vasa proteins remain on-site on the spindle. As another negative control, we analyzed the translation initiation of TC-Vasa-ΔC-term-GFP (103 aa of Vasa C-terminal deletion) that lacks both the spindle-associated localization and the function as a cell cycle factor[19]. In these embryos, both the GFP and ReAsH signals were

observed in the cytoplasm and less on the spindle (Fig. 4h and Supplementary Fig. S6c, d), suggesting the site of translation differs for non-spindle associated proteins. Taken together, we conclude that Vasa mRNA is locally translated and remains on the spindle, potentially for its immediate role in translation regulation of other mRNAs on the spindle.

The above results also suggest that Vasa's C-terminal Region (Fig. 5a, pink region) contains an essential region for Vasa's activity in localized translation on the spindle. In the past, various Vasa mutations especially in the middle DEAD-box helicase region (Fig. 5a, blue region) were reported to be important for Vasa's function in the insect germline[17,18,36]. Further, a more recent report suggests that the C-terminal region of Vasa is important for its function in piRNA biogenesis of the *Drosophila* germline[37]. These observations suggest that Vasa has multiple functional domains to accommodate its multi-functionality in the cells. To test what region(s) of Vasa is essential for its function in localized translation on the spindle, a series of Vasa deletion or point mutation constructs were tested (Fig. 5a and Supplementary Fig. S7a). Each of the Vasa mutants' localization was first analyzed by a GFP reporter (Fig. 5b and Supplementary Fig. S7b). All of the known point mutations of Vasa showed little effect on Vasa's localization on the spindle. On the other hand, the Vasa C-terminal Region was consistently found to be important. Especially, its last three amino acids (E-S-W-D) appear to be critical for Vasa's spindle localization (Fig. 5b, Vasa-C3). Notably, knocking down endogenous Vasa translation with a Vasa Morpholino antisense oligo (Vasa-MO) results in cessation of most translation and cell cycle activity, which is rescued by the introduction of Vasa-GFP[22]. A series of Vasa mutants that included mutations in the same last three amino acids (Vasa-C3 and Vasa-C1–3) showed compromised translation activities and poor rescue, equal to negative controls (Vasa-MO, Vasa-C1–3, Vasa-C3 groups; Fig. 5c, d and Supplementary Fig. S7c–e). These results suggest that the three amino acids at the C-terminus are critical both for Vasa's localization and translation activity on the spindle. Importantly, the sequence of these C-terminal three amino acids of Vasa is highly conserved among various organisms, including flies, mice, humans, yet distinct from the rest of the DEAD-box helicases[37,38]. This suggests that Vasa's function on the spindle may be conserved among organisms yet molecularly unique among DEAD-box helicase family proteins. We conclude that the last three amino acids of the Vasa C-terminus are essential for Vasa's functions in localization on the spindle, cell cycle progression, and general translation.

**Ectopic Vasa function induces developmental failure**. The above results demonstrate that asymmetric Vasa enrichment on

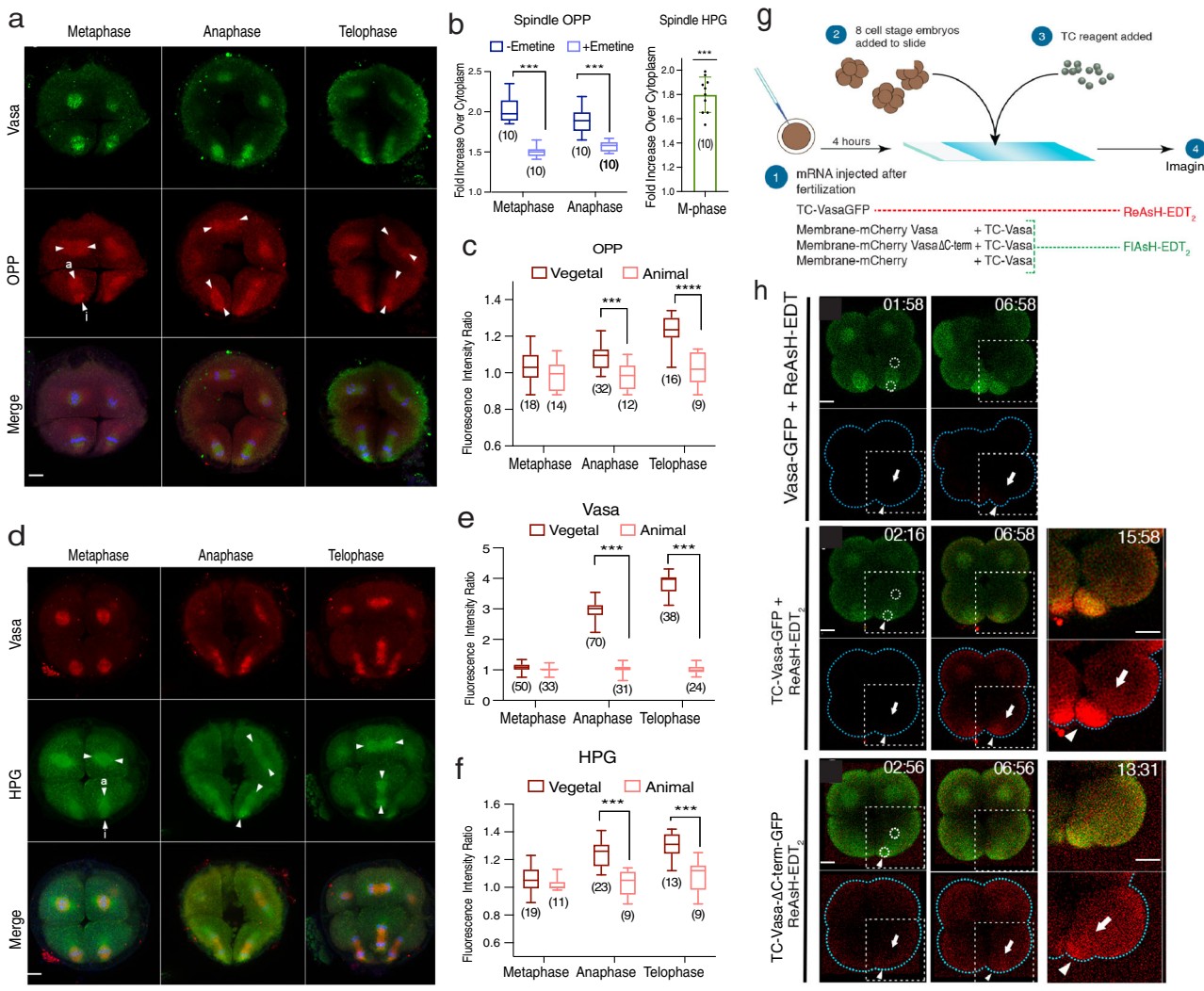

**Fig. 4 In vivo or real-time detection of localized translation. a** The OPP (red) and Vasa (green) staining at the 16-cell stage. i and e arrowheads indicate the micromere and macromere sides of the spindle areas, respectively, analyzed in (**c**). A whole Z-section of the cell (1 μm per slice; ~35 slices in total) presented as a maximum 2D-projection. **b** (Left) % of the OPP signal on the spindle area with/without emetine (0.5 μM). (Right) % of the HPG signal on the spindle. $n = 10$ for all groups. Adjusted $p$-value <0.0001. **c** The OPP level comparison between the micromere side (i) and the macromere side (**a**) of the spindle at telophase. $n = 18, 14, 32, 12, 16, 9$ from the left-right columns. **d**–**f** Vasa (red; **e**) and HPG (green; **f**) signals enriched on the micromere side (i) of the spindle compared to the macromere side (**a**). $n = 50, 33, 70, 31, 38, 24$ from the left-right columns of graph (**e**). $n = 19, 11, 23, 9, 13, 9$ from the left-right columns of graph (**f**). Adjusted $p$-value <0.0001. A whole Z-section of the cell (1 μm per slice; ~35 slices in total) presented as a maximum 2D-projection. **g** TC-tag imaging procedure. **h** Live imaging of TC-tagged GFP (green) during micromere formation in 3 μM ReAsH-EDT$_2$ (red): Vasa-GFP with no TC-tag (negative control), TC-Vasa-GFP (wild type), and TC-Vasa-ΔC-term-GFP (non-functional mutant). Time codes are the time post ReAsH-EDT$_2$ exposure. A sub-Z-section (~5 μm) of the embryo around the spindle area was imaged every 15-s. All experiments were repeated at least three independent times. () in all graphs indicate the total number of embryos analyzed. The box plot defined by two box lines indicates the 25th and 75th percentile, respectively, with a centerline at the 50th percentile and the minima and maxima whiskers. ***$p$ < 0.001, ****$p$ < 0.0001. Columns represent means ± SD or SEM. Two-way ANOVA was used for all statistical analyses except for the Spindle HPG (**b**, right) that used paired $t$-test in this figure. Source data are provided as a Source Data file. Scale bars = 10 μm.

the spindle results in asymmetric protein synthesis over the spindle. We hypothesize that this positive reinforcing cycle is responsible for the increased translation of various mRNAs, including *vasa* mRNA itself. To determine how Vasa accomplishes this function, we tested its ability to recruit other factors essential for translation, and even to nucleate translation activity. We first targeted Vasa to the plasma membrane by fusing it to membrane-mCherry (Fig. 6a and Supplementary Fig. S8a). In these embryos, even though endogenous Vasa was still present in the cell, Vasa mRNAs, Vasa proteins, and general protein synthesis (HPG signals) were all enriched at the plasma membrane and reduced at the spindle, suggesting that a significant

part of the translation machinery was sequestered to the plasma membrane (Fig. 6b–e and Supplementary Fig. S8b–e, arrowheads). As negative controls, membrane-mCherry, membrane-mCherry-Vasa-ΔC-term (Supplementary Fig. S8), or membrane-mCherry-Vasa-C3 (Fig. 6b–e) were introduced into the embryo. In all negative control groups, Vasa mRNA or HPG signal was always enriched on the spindle but never enriched on the plasma membrane, suggesting no ectopic translation activity at the plasma membrane. Importantly, embryos expressing membrane-targeted Vasa showed abnormal cell division often accompanied by the lack of micromeres, and failure in gastrulation with no Vasa enrichment within a germline (Fig. 6f and Supplementary Fig. S8f).

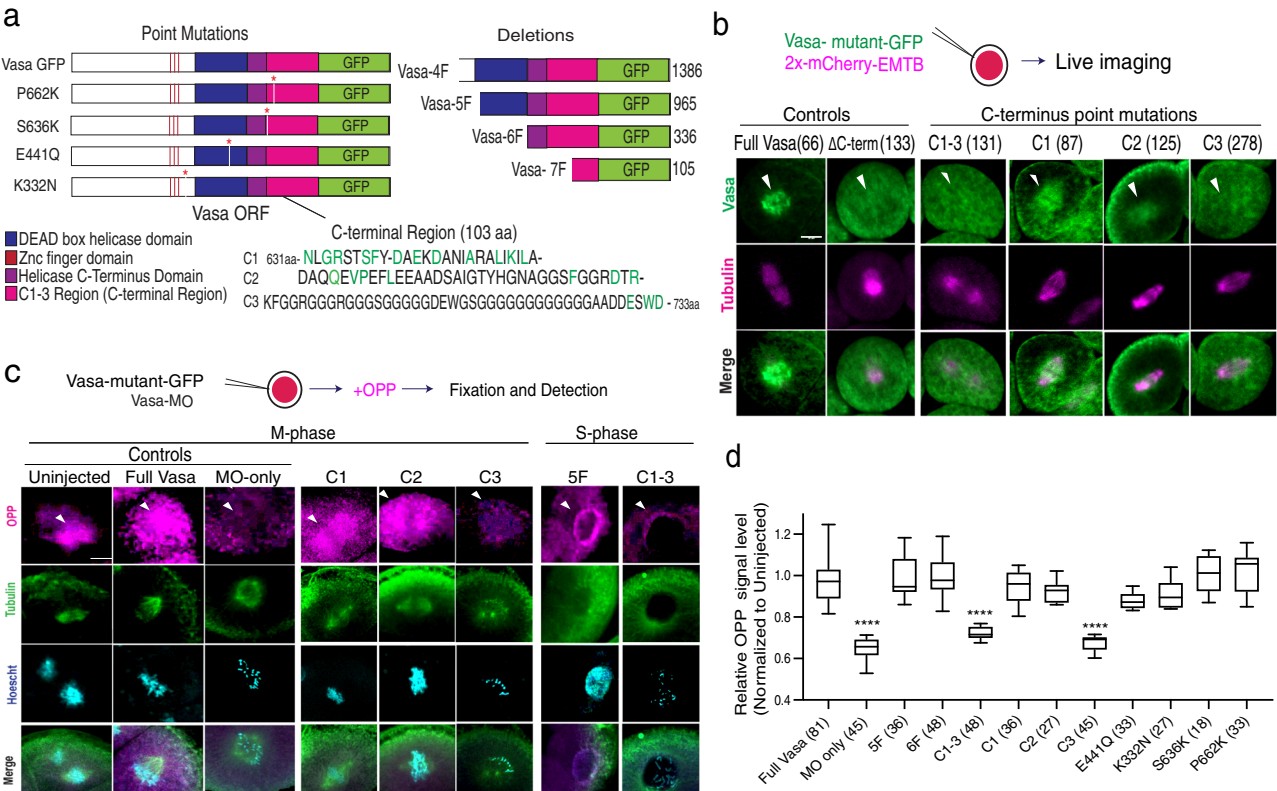

**Fig. 5 Vasa C-terminus is responsible for localized translation on the spindle. a** A schematic diagram of Vasa deletion or mutation constructs tested in this study. VasaC1-3 series are the constructs mutated for the Vasa-C-terminal regions (see also Supplementary Fig. S7 for detail). **b** Live imaging of the 8–16-cell embryonic cell injected with each of the Vasa mutants fused to GFP. Vasa's spindle localization was lost in Vasa-C-terminal mutants (e.g., Vasa-ΔC-term, Vasa-C1-3, and Vasa-C3), yet not in other mutants except for Vasa-7F that lacks all of the conserved motifs important for DEAD-box helicase activities (Supplementary Fig. S7). **c** Localized translation on the spindle was measured by the OPP signal level in each embryo injected with each of the Vasa mutants along with Vasa morpholino antisense oligo to knock down endogenous Vasa activity. The OPP signal appears on the spindle when the introduced Vasa mutants are functional. A whole Z-section of the cell of interest (1 μm per slice; ~35 slices in total) was imaged, analyzed, and presented as a maximum 2D-projection. The whole embryo images are presented in Supplementary Fig. S7. **d** Vasa C-terminus deletion or mutation resulted in the reduced signal of localized translation on the spindle. The relative OPP intensity level on the spindle versus in the cytoplasm was obtained as follows: The average signal level of three detection ROIs was obtained per embryo. A total of ~27 embryos were analyzed in this manner per sample group and the average value was presented in the graph. ( ) indicates the total number of ROIs measured for each sample group. All experiments were performed at least three independent times. Adjusted *p*-value <0.0001. *n* = 81, 45, 36, 48, 48, 36, 27, 45, 33, 27, 18, 33 from the left-right columns of graph (**d**). The box plot defined by two box lines indicates the 25th and 75th percentile, respectively, with a centerline at the 50th percentile and minima and maxima whiskers. Two-way ANOVA was used for all statistical analyses in this figure. ****p* < 0.0001. Columns represent means ± SD or SEM. All scale bars = 10 μm. Source data are provided as a Source Data file.

This developmental defect was surprising to us because over-expression of full-length or any mutants of Vasa never shows developmental defect nor function as a dominant-negative in embryos of any organisms, especially in somatic lineages[39]. For instance, in sea urchin embryos, overexpression of Vasa mRNAs results in only inactive degradation of excessive Vasa proteins both in the cytoplasm and on the spindle, resulting in no overall increase in Vasa protein nor developmental defect[19].

These observations further reinforce the idea of where Vasa functions within the cell are more critical than the amount of its expression in the cell. We propose that membrane-targeted Vasa sequesters translation factors and/or mRNAs to the plasma membrane, which dislocates the factors essential for cell divisions and embryonic cell fate determination from their normal position, causing cellular and developmental failure. Membrane-mCherry-Vasa-ΔC-term or -Vasa-C3, on the other hand, lacks these recruitment activities and thus are innocuous for development. These results support the contention that proper sub-cellular function of Vasa, i.e., sites of protein synthesis, is essential for proper embryonic development. Further, the last three amino

acids of <u>E</u>-<u>S</u>-<u>W</u>-<u>D</u> are critical for Vasa's function in localized translation on the spindle.

**Plasma membrane-targeted Vasa nucleates ectopic translation.** We next tested how membrane-targeted Vasa facilitates ectopic translation using the TC-FlAsH (green) detection system described above. TC-Vasa (non-fluorescent) was co-expressed with membrane-mCherry-Vasa (red), and FlAsH reagent was added at the 8–16 cell stage. The resulting red and green signal showed a higher degree of overlap in the embryos injected with membrane-mCherry-Vasa and TC-Vasa compared to the control group (membrane-mCherry-Vasa-ΔC-term and membrane-mCherry groups; Supplementary Fig. S9a–c). For red fluorescence (mCherry), the peak intensity occurs at the cortex, so overlap of red and green peaks indicate correlative membrane-bound Vasa protein and translation activity (Supplementary Fig. S9b). For membrane-mCherry-Vasa embryos, red and green peaks overlapped completely or nearly (0~0.2 μm) at 2:55 min after TC reagent addition. On the other hand, membrane-mCherry-Vasa-ΔC-term or membrane-mCherry alone embryos

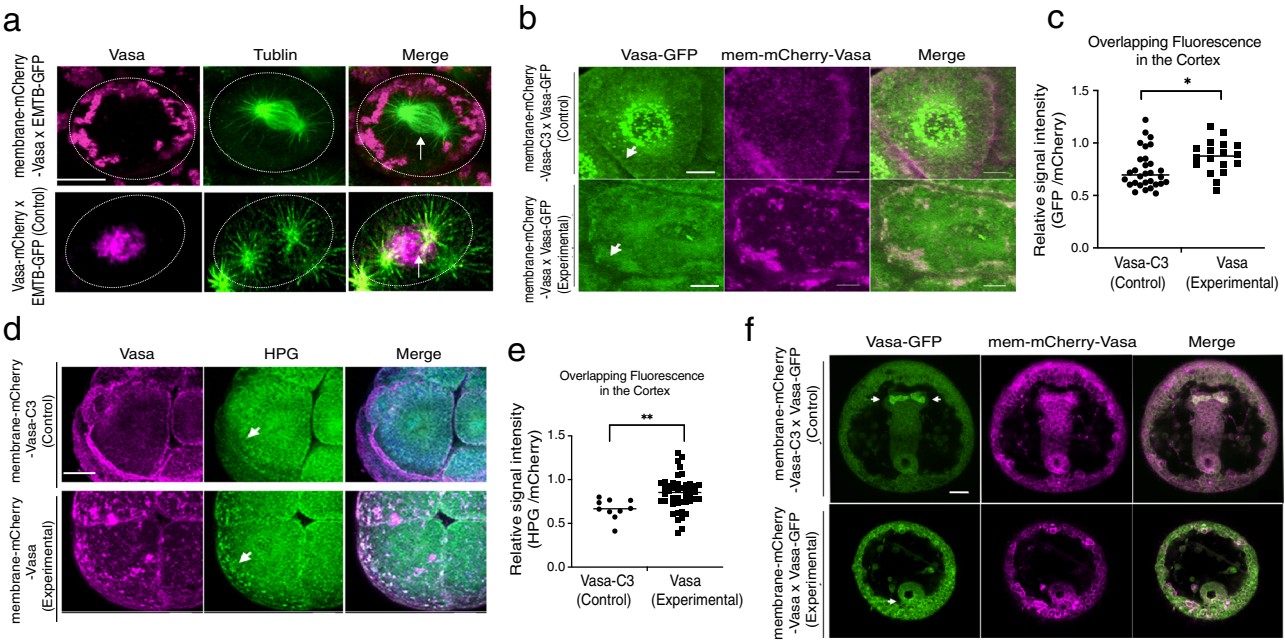

**Fig. 6 Vasa nucleates ectopic translation on the membrane. a** membrane-mCherry-Vasa (magenta, upper panels) was recruited to the membrane (outlined by a white circle) and away from the spindle (arrow) (n = 10), while Vasa-mCherry (magenta; control; lower panels) was enriched specifically on the spindle (n = 10). EMTB-3xGFP (green) was used as a microtubule marker. **b** Vasa-GFP (green) mRNA was coinjected with membrane-mCherry-Vasa or Vasa-C3 (red) mRNA. The overlapping signal of GFP and mCherry is shown in orange. membrane-mCherry-Vasa, n = 33; membrane-mCherry-Vasa-C3 (n = 29). **c** The graph represents the overlapping mean fluorescence on the cortex (green/ magenta) of controls versus experimental conditions. Average values from 3 detection ROIs per embryo were obtained for each group, n = 30 for membrane-mCherry-Vasa-C3 (a total of 90 ROIs) and n = 18 for membrane-mCherry-Vasa (a total of 54 ROIs). Adjusted p-value, 0.022. **d** New protein synthesis (HPG signal, green) was detected on the plasma membrane with membrane-mCherry-Vasa (arrows, n = 18), but little with membrane-mCherry-Vasa-C3 (n = 30) in 16-cell stage embryos. A whole Z-section of the cell of interest (1 μm per slice; ~35 slices in total) presented as a maximum 2D projection. **e** The graph represents the overlapping mean fluorescence on the cortex (green/ magenta) of the control versus experimental condition. Average values from 3 detection ROIs per embryo were obtained for each group, n = 10 for membrane-mCherry-Vasa-C3 (a total of 30 ROIs) and n = 45 for membrane-mCherry-Vasa (a total of 135 ROIs). Adjusted p-value, 0.009. **f** Membrane-mCherry-Vasa injected embryos failed in gastrulation as well as Vasa localization in the germline (arrows). Membrane-mCherry (magenta), Vasa-GFP (green). A whole Z-section of the cell of interest (1 um per slice; ~35 slices in total) presented as a maximum 2D projection. All experiments were performed at least three independent times. All scale bars except for (**f**) = 10 μm; Scale bar of (**f**) = 20 μm. Multiple t-tests were used for all statistical analyses in this figure. *p < 0.05, **p < 0.01. Columns represent means ± SD or SEM. Source data are provided as a Source Data file.

showed 0.7 ~ 17 μm or 0.8 ~ 12 μm distance between peaks - at 2:55, respectively (Supplementary Fig. S9d). After 5 min, the translated proteins appeared to depart from the cortex yet remained in close proximity, implying the site of mRNA translation is closely linked to its corresponding protein localization, and potentially to its immediate function. Further, we found that a general translation factor eEF1A that is responsible for the selection and binding of the cognate aminoacyl-tRNA to the A-site of the ribosome[40], interacts with Vasa, and is enriched on the spindle of control embryos (Supplementary Fig. S9e–g). In the experimental embryos injected with membrane-targeted Vasa, on the other hand, some of eEF1A proteins were co-localized on the cortex with membrane-targeted Vasa (Supplementary Fig. S9g). These results support the contention that Vasa nucleates translation by recruiting translation machinery and mRNAs, and that competition for limiting translation machinery from the spindle results in developmental defects.

**Vasa asymmetry is essential for cell fate regulation.** To test if asymmetric Vasa on the spindle contributes to asymmetric translation on the spindle and cell fate determination during asymmetric cell division, we employed a LOV-ePDZ mediated optogenetic system. The photosensitive LOVpep (LOV) domain from *Avena sativa* phototropin1 undergoes a conformational change and binds to the engineered PDZb1 (ePDZ) domain upon

blue light irradiation[41–43]. This binding is reversible and can be released when blue light irradiation is ceased or significantly decreased, providing good temporal control of protein-protein binding[44]. To recruit Vasa to an ectopic sub-cellular region, Vasa-mCherry was tagged to LOV. Lifeact (actin-binding domain)[45] was tagged to ePDZ, which recruited a significant portion of Vasa-mCherry-LOV protein away from the spindle upon blue laser irradiation (Supplementary Fig. S10a–c). To avoid unwanted optogenetic activation while imaging, a minimum amount of laser power (0.1~0.3% for 488 nm and 1~3% for 561 nm) was also used. These conditions were previously documented to induce Vasa-less micromeres in this embryo[44]. In these embryos with optogenetics cassettes, translation activity (detected by FIAsH; green) dispersed from the spindle to the cytoplasm upon blue light irradiation, resulting in no FIAsH enrichment in micromeres. On the other hand, the control embryos that were injected with the same cassettes but without the *LOV domain* showed FIAsH enrichment in the micromeres (Fig. 7a–d and Supplementary Fig. S10d–f; Movies S8–9). This result suggests that Vasa's asymmetric localization is essential for the asymmetric translation of mRNAs on the spindle.

Further, we tracked the cell fate of these embryos, using a polarity marker GFP-β-catenin or a germline marker Nanos-GFP. In normal development, nuclear β-catenin is localized to the most vegetal cells and shows a gradient diminishing toward the animal

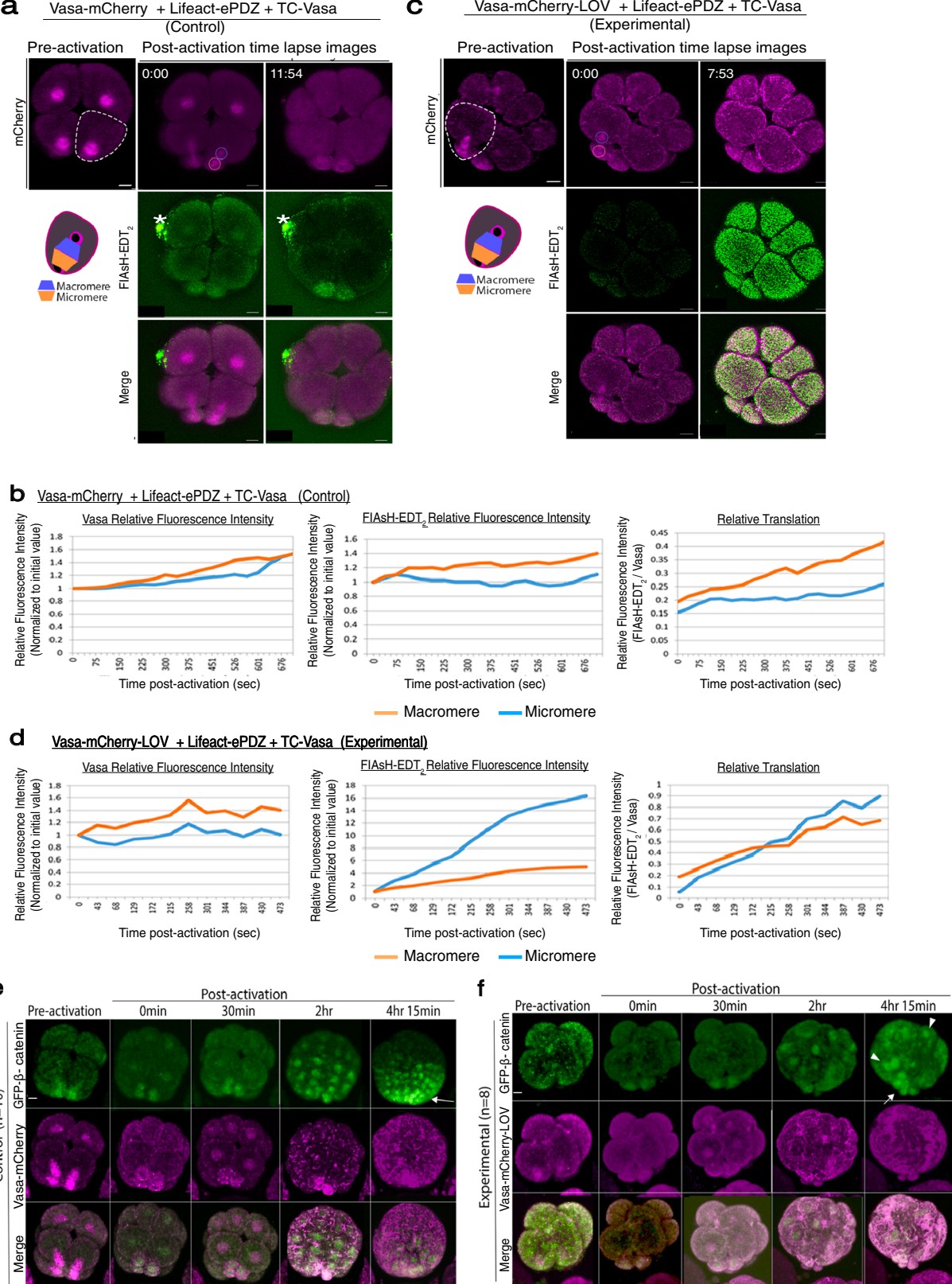

pole, assisting the establishment of the animal-vegetal axis in the embryo[46,47]. Nanos becomes enriched into the germline during early embryogenesis (5~36 hpf), contributing to germline specification[48]. In the light-activated embryos, however, proper localization of both nuclear β-catenin and Nanos was altered

(Fig. 7e, f and Supplementary Fig. S11a, b; Movies S10–11). On the other hand, no disruption of the nuclear β-catenin gradient was ever observed when the Vasa-C3 mutant was recruited to the membrane even in a constitutive manner (Supplementary Fig. S11c). These results further suggest that Vasa's specific and

**Fig. 7 Optogenetic disruption of Vasa localization on the spindle during asymmetric cell division altered localized translation and cell fate of the micromeres. a–d** Upon blue light irradiation, Vasa-mCherry-LOV was recruited to actin, resulting in Vasa-less micromeres at the 16-cell stage in the experimental group. A level of translation was detected by FlAsH-EDT$_2$ at the 16-cell stage and was increased in the micromere-side of the spindle in the control group (**a** and **b**) whereas decreased in that of the experimental group (**c** and **d**). A whole Z-section of the cell of interest (0.44 µm per slice; ~90 slices in total) was imaged and analyzed every 1-min and presented as a maximum 2D-projection. A bright green spot (*) is a non-specific debris that tends to fluoresce under the microscope. Each experiment was repeated at least three times and the measurement results of two other embryos are shown in Supplementary Fig. S10f. **e, f** Localization of β-catenin-GFP was tracked by time-lapse imaging of every 15 min for ~5 h after activation (approximately, ~12 hpf). In controls, GFP-β-catenin enriched into the vegetal most cells (**e**, arrows), whereas its localization became uniform or random in the experimental groups (**f**, arrows). Arrowheads indicate ectopic localization of GFP-β-catenin. A whole Z-section of the embryo of interest (1 µm per slice; ~50 slices in total) was imaged and is presented as a maximum 2D-projection. The embryos were lightly squashed to completely immobilize during imaging. Therefore, the embryos that successfully formed micromeres under this condition were selected for the analysis and the embryos in a similar angle were compared between control and experimental groups in this study. These experiments were performed at least three independent times. All Scale bars = 10 µm. Source data are provided as a Source Data file.

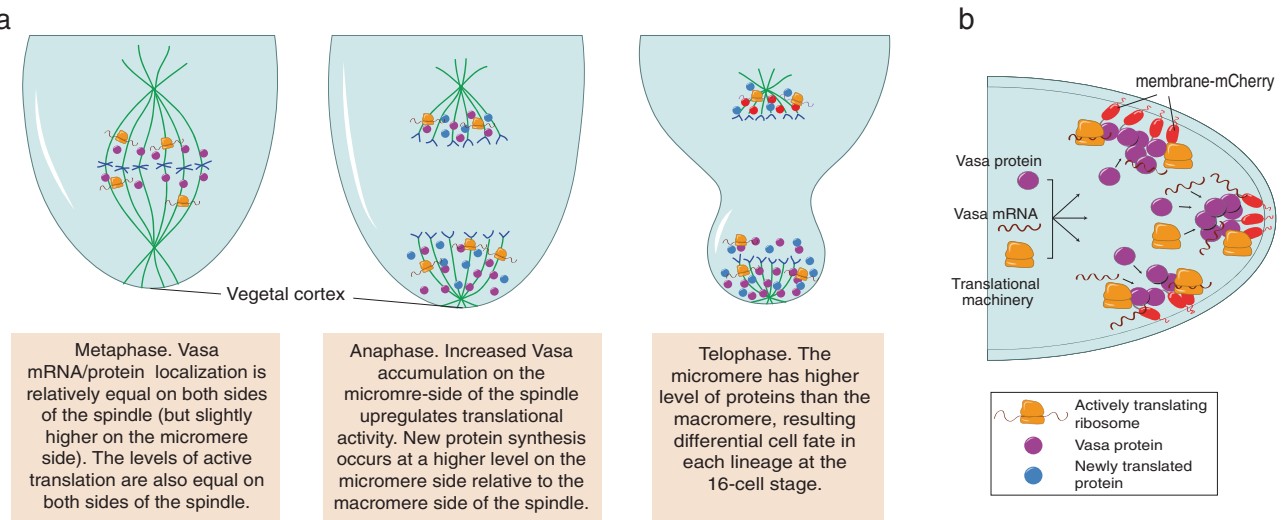

**Fig. 8 Models for Vasa enrichment and new protein synthesis on the mitotic spindle. a** A model for asymmetric translation on the spindle during micromere formation. Metaphase: Vasa (purple) is equally abundant on both sides of the spindle, so as are levels of active translation. Anaphase: Increased active translation on the micromere side of the spindle facilitates increased Vasa accumulation in the same region. Increased Vasa also increases translation activity, and new protein synthesis occurs at a higher level on the micromere side relative to the macromere side. Telophase: the micromere has a higher level of protein synthesis, resulting in the micromere-specific fate differentiation. **b** A model for ectopic translation nucleated by membrane-targeted Vasa. Membrane-targeted Vasa recruits Vasa protein, mRNAs, and other spindle-associated molecules as well as translation machinery for ectopic on-site translation on the plasm membrane, which results in cell division and developmental defects.

asymmetric activity on the spindle is critical for the proper molecular organization, which is essential for cell fate specification and inductive-based embryonic development.

**Vasa functions in localized translation on the spindle**. In summary, we conclude that Vasa is essential for localized translation in the sea urchin embryo. Vasa also increases general protein synthesis asymmetrically for efficient cell fate differentiation (Fig. 8a, b). Considering that overexpression of Vasa generally induces no developmental defects in various organisms, its essential role might be in recruiting and assembling translation machinery and its target mRNAs to a specific sub-cellular region. This might allow a cell to facilitate efficient mRNA translation with temporal and spatial control during rapid cellular development. Indeed, essential Vasa function has been also reported in cancer cells, stem cells, and regenerative tissues, each of which undergoes rapid cell proliferation and dynamic gene and/or protein expressions[39,49–51]. Further, localized translation on the mitotic apparatus has been also proposed in various cell types such as neuronal cells, cancer cells, *Xenopus* embryos, and mouse embryos for the last decades[3]. Therefore, the fundamental role of Vasa in localized translation might be widely conserved among

various cell types and organisms, serving as an essential mechanism for cellular and embryonic survivability.

## Methods

**Sea urchin embryo culture**. *Strongylocentrotus purpuratus* (sea urchins) were collected from the ocean by Pat Leahy, Kerchoff Marine Laboratories, California Institute of Technology, or by Josh Ross, South Coast Bio- Marine LLC. Long Beach, California, USA, and maintained in the cooling aquarium at 16 C. Sea urchin eggs were collected in seawater (SW) and sperm was collected dry through 0.5 M KCl injection to induce spawning. Eggs and embryos were cultured in SW at 16 °C. For fixation, fertilization was performed in the presence of 1 mM 3-amino triazole (Sigma, St. Louis, MO, USA) to remove fertilization envelopes.

**Generation of plasmid constructs**. Kaede was selected as a photoconversion tool in this paper because it outlined the overall structure of the spindle including asters during recording. Further, it has been reported that Vasa mRNA overexpression causes no developmental defect[19,22], and thus Kaede-Vasa served as a useful tool to track the protein dynamics. Kaede was amplified by PCR from the Kaede-centrin1 template[52] and subcloned into the pSP64-vector to construct pSP64-Kaede. Or, Kaede was fused to the N-terminus of Vasa ORF or Vasa1F3R (Vasa-ΔC-term) that lacks 103aa of Vasa C-terminal region, spindle-associated localization, and function as a cell cycle factor[19,22], and subcloned into the pSP64-vector, containing the *Xenopus* β-globin 5′ and 3′ UTRs as well as an Sp6 promoter and transcription start site, to construct pSP64-Kaede-Vasa or pSP64-Kaede-VasaDead, respectively. A series of Vasa deletion constructs were previously used in Yajima and Wessel, 2015. Vasa-point mutation constructs were made by inserting synthesized DNA

**Table 1 A list of all materials used in this study.**

| Reagent or resource | Source | Identifier |
| --- | --- | --- |
| Antibodies | | |
| Anti-SpVasa | Voronina et al., 2008[54] | N/A |
| Anti-EF1A | Abcam | # ab175274 |
| Anti-RPS6 | Cell signaling technology | # 4858 |
| FITC-conjugated Anti-beta-tubulin | Sigma-Aldrich | # F2043 |
| Anti-mCherry | Thermo Fisher Scientific | # PA5-34974 |
| Cy3-conjugated goat anti-rabbit IgG | Thermo Fisher Scientific | # A10520 |
| Alexa 488-conjugated goat anti-mouse IgG | Thermo Fisher Scientific | # A32723 |
| HRP-conjugated goat anti-rabbit IgG | Thermo Fisher Scientific | # A16096 |
| Chemicals, Peptides, and Recombinant Proteins | | |
| Hoechst 33342 | Thermo Fisher Scientific | # 62249 |
| Click-iT® Plus OPP Protein Synthesis Assay Kits | Life technologies | # C10457 |
| Click-iT® HPG Alexa Fluor® Protein Synthesis Assay Kits | Life technologies | # C10428 |
| Red fluorescent dextran (Fluoro-Ruby) | Invitrogen | # D1817 |
| Emetine dihydrochloride hydrate | Sigma-Aldrich | # E2375 |
| TC-FlAsH™ TC-ReAsH™ II In-Cell Tetracysteine Tag Detection Kits | Molecular Probes | # MP34561 |
| Critical Commercial Assays | | |
| mMESSAGE mMACHINE SP6 Transcription Kit | Ambion | # AM1340 |
| DIG RNA Labeling Kit (T7) | Roche | # 11 175 025 910 |
| In-Fusion HD Cloning | Clonetech | # 639648 |
| Recombinant DNA | | |
| Plasmid: Vasa-GFP | Yajima and Wessel, 2011b | N/A |
| Plasmid: Kaede/Dendra-Vasa | This study | N/A |
| Plasmid: 2xmCherry-EMTB | Addgene | # 26742 |
| Plasmid: EMTB-3xGFP | Addgene | # 26741 |
| Plasmid: Vasa deletion/mutation series | This article | N/A |
| Plasmid: membrane-mCherry-Vasa/Vasa-ΔC-term | This article | N/A |
| Plasmid: TC-Vasa/Vasa-ΔC-term -GFP | This article | N/A |
| Plasmid: Lifeact/Lifeact-GFP-ePDZ | Uchida and Yajima, 2018[44] | N/A |
| Plasmid: Vasa-mCherry-LOV | This article | N/A |
| Plasmid: GFP-β-catenin | This article | N/A |
| Plasmid: Nanos-GFP | Oulhen et al., 2010 [53] | N/A |
| Sequence-Based Reagents | | |
| Morpholino antisense oligos | Gene tools | http://www.gene-tools.com/ |
| Software and Algorithms | | |
| Echinoderm gene/protein sequences | EchinoBase | http://www.echinobase.org/Echinobase/ |
| Imaging software | Olympus Fluoview and CellSense Or Zeiss Zenblue NIS Elements | |
| Quantitative analysis | *Image J* | https://imagej.nih.gov/ij/ |
| Statistical analysis | PRISM | https://www.graphpad.com/scientific-software/prism/ |

fragments containing each mutation (IDT, Illinois, USA) into the pSP64-Vasa-ΔC-term -GFP vector. All the subcloning reactions were performed using the In-Fusion HD Cloning kit by following the manufacturer's protocol (#639648, Clontech, USA). Similarly, Dendra2 (Addgene #54694) was subcloned into pSP64-vector and GFP-Spβ-catenin (Logan and McClay, 1999; SPU_009155) was synthesized by IDT (Illinois, USA) and inserted into the same vector as described above. pCS2-2xmCherry-EMTB and pCS2-EMTB-3xGFP[20] were obtained at Addgene (#26741). pSP64-SpNanos-GFP was previously made[53].

To make TC tagged proteins, site-directed mutagenesis primers containing the TC tag sequence TGTTGTCCTGGCTGTTGC were designed using the QuikChange Primer Design Tool (Agilent, USA) [or NEBase Changer v1.2.4, (New England Biolabs, USA)]. The QuikChange II Site-Directed Mutagenesis Kit (#200523, Agilent, USA) [or Q5® Side-Directed Mutagenesis Kit (#E0554, NEB, USA)] was used to insert the TC tag at the N-terminus of the protein of interest into existing plasmid templates. Mutagenesis was used to generate pSP64-TC-Vasa-GFP, pSP64-TC-Vasa.

Membrane-mCherry was amplified by PCR and inserted into the pSP64-Vasa ORF or pSP64-Vasa-ΔC-term vectors described above, using an In-Fusion HD Cloning kit to produce pSP64 Membrane-mCherry-Vasa and pSP64 Membrane-mCherry-Vasa-ΔC-term /C3. pSP64-Membrane-mCherry was constructed by cutting template plasmid pSP64 Membrane-mCherry-PLK1 ORF[32] with SpeI to drop out the PLK1 ORF, gel purifying the resulting pSP64-Membrane-mCherry fragment, and re-ligating with T4 DNA ligase (#M1801, Promega, USA) according

to the Promega protocol. Please also see Table 1 for a list of all resources used in this study.

**Morpholino design**. Morpholino antisense oligos (MOs) specific to SpVasa and SpNanos (used as a control in this work) were previously used and its specificity has been confirmed[19]. 1.5 mM stock of SpVasa was used for injection.

**RNA injection into sea urchin zygotes**. Plasmids were linearized with SalI (Promega) and transcribed in vitro with SP6 mMessage machine kit (Ambion, USA). RNA product was resuspended in nuclease-free water at the stock concentration of 0.5~2 μg/μl. Fertilized eggs were injected with approximately 6~20 pl each of the stock RNA solution and incubated at 16 °C until they were imaged at the 8-cell stage for the sea urchin embryos.

**Embryo imaging**. For time-lapse imaging, Live embryos were placed on a cooling stage at 16 °C (Linkam PE100, McCrone, UK) to be visualized with an Olympus FV3000 or Nikon CSU-W1 spinning disk confocal laser scanning microscope that minimized cell toxicity and photo-bleaching.

For Kaede-Vasa photoconversion experiments, Kaede-Vasa on the spindle of either animal or vegetal blastomere was irradiated with 1% 405 nm laser for 0.25 s and imaged subsequently on the Olympus FV3000 point scanning confocal

microscope with a resonant scanner. Although it penetrates the entire Z-planes, the corn-shaped laser was designed to hit the most at the specific Z-plane of interest (region of interest, ROI), using the short and minimum laser power irradiation. This allowed only the ROI to reach the threshold for visible photoconversion yet not anywhere in the cytoplasm, which was confirmed by the 3D rotation of the image immediately after each photoconversion (Supplementary Fig. S2c; Movie S3). Further, this experimental condition showed no noticeable cell cycle delay or defect in the irradiated cell compared to the sibling blastomeres within the same embryo. We also tested the multi-photon imaging approach, which required significantly higher power lasers both for photoactivation and imaging, and immediately caused abnormal cell blebbing and cell cycle delay in the embryo[23]. In this study, therefore, we employed the former approach with a resonant scanner to minimize the unwanted photoconversion outside of the ROI as well as developmental toxicity.

For optogenetic experiments, embryos injected with Vasa-mCherry-LOV and Lifeact-GFP-ePDZ/Lifeact-ePDZ were irradiated with 488 nm for 0.5~0.8 sec. Each Z-stack series was taken every 15~60 s for 10~60 min for Kaede-Vasa and FlAsH tracking, or every 15 min for ~7 h for Nanos- and β-catenin-GFP tracking. Fluoview or *Image J* (NIH) was used for image processing and fluorescence intensity measurements. For still imaging with fixed samples, embryos were mounted on glass slides in 1X PBS, then imaged by confocal laser microscopy (Zeiss LSM800, Olympus FV3000, and Nikon CSU-W1 spinning disk) and processed by the software attached to each microscope such as Olympus Fluoview (Version 4.2), Zeiss Zen (blue edition) and Nikon NIS Elements (Advanced Research Package).

**In vivo protein detection and chemical treatment.** HPG is a glycine derivative and is incorporated during new protein synthesis and thus works as a vital method for visualizing nascent protein populations synthesized at specific times as determined by the reagent exposure. O-propargyl-puromycin (OPP) is a puromycin derivative, distinct from HPG, but is also incorporated in vivo into an active translational complex. OPP reagent (20 mM stock solution, Life Technologies) was added to each subset of embryos at a 1:1000 dilution for a final concentration of 20 μM at the 8-cell stage, approximately 4.5 h after fertilization. HPG reagent (50 mM stock solution, Life Technologies) was added to another subset of developing embryos at a 1:1000 dilution for a final concentration of 50 μM at the 2-cell stage, approximately 2 h after fertilization. The remaining third received only HPG treatment. As a negative control, another subset of embryos with no treatment was prepared and subjected to detection. For emetine (emetine dihydrochloride hydrate, Sigma-Aldrich; *52, 53*) treated embryos, OPP or HPG reagents were added as described above, and then emetine was added to the incubation media 30 min prior to fixation for a final concentration of 0.5 mM. In this condition, translation was partially blocked, which allowed embryos to continue cell cycling until they reached the desired stage for sample collection. Pretreatment or longer incubation of emetine, on the other hand, halted cell cycling. For detection, embryos were fixed at anaphase of the fourth mitotic cell division approximately 5 h after fertilization with 90% methanol (MeOH) and incubated at −20 °C overnight. They were then washed once with 1X phosphate-buffered saline pH 7.5 (PBS) and the Click-iT detection system was applied to these embryos according to the Life Technologies kit protocol. They were then washed once with 1X PBS before immunofluorescent staining. All HPG/OPP signal quantification was conducted on *ImageJ*. Please also see Table 1 for a list of all resources used in this study.

**Immunofluorescence, immunoprecipitation & blotting, and in situ RNA hybridization.** HPG and OPP treated embryos were stained with a primary antibody against Vasa[54] diluted 1X PBS at 1:300 or 1:200 at room temperature, overnight. They were washed 6 times with 1X PBS and stained with a secondary antibody against rabbit (1:500, 1 mg/mL, Invitrogen) in 1X PBS. They were then washed 6 times and a Hoechst nuclear stain (10 mg/mL, Promega) in 1X PBS was applied at a 1:1500 dilution.

Untreated embryos fixed at anaphase with the same protocol used for HPG and OPP treated embryos, without Click-iT detection system application: Fixed embryos were stained with primary antibodies against Vasa[54] at 1:300, EF1A (#ab175274, Abcam, USA) at 1:100, DDX4/MVH antibody (#ab13840, Abcam, USA) at 1:100, and RPS6 (Cell signaling #4858) that recognizes ribosomal protein S6 when it is phosphorylated at Ser235 and 236 and is a commonly used marker of active translation (9) at 1:100. They were washed 6 times with 1X PBS and stained with a secondary antibody against rabbit (1:500) and goat (1:500) (1 mg/mL, Invitrogen). They were washed and stained with Hoechst as stated above.

For immunoprecipitation and immunoblotting, approximately 1 μg of mCherry antibody (#PIPA534974, Thermo Scientific, USA) per 300 embryos injected with membrane-mCherry-Vasa or membrane-mCherry was used for immunoprecipitation (IP). The IP was performed as described in the instruction manual of Dynabeads Protein A (Invitrogen). The resultant samples were then prepared in 50 μL of loading buffer for polyacrylamide gel electrophoresis. Each sample (15 μl) was run on a 4–20% gradient Tris-glycine polyacrylamide gel (Invitrogen, Carlsbad, CA) and transferred to nitrocellulose membranes for immunoblotting with mCherry antibody or EF1A antibody at 1:1000 for mCherry-IP, and with peroxidase-conjugated anti-mouse or -rabbit secondary antibodies at 1:5000 (Life Technologies), respectively. The reacted proteins were detected by

incubation in a chemiluminescence solution (1.25 mM luminol, 68 μM coumaric acid, 0.0093% hydrogen peroxide, and 0.1 M Tris pH8.6) for 1–10 min, exposed to film, and developed. Each experiment was performed at least three independent times.

For in situ hybridization, the RNA probe for *Sp-vasa* was previously constructed and prepared as described in Yajima and Wessel, 2014 & 2015. Briefly, a DIG RNA Labeling Kit (T7) (Roche, Indianapolis) was used to construct antisense DIG-labeled probes to each target mRNA from a cDNA template, and the DIG-labeled RNAs were then hybridized to embryos as described previously[22,55]. Please also see Table 1 for a list of all resources used in this study.

**Optogenetic activation and imaging.** mRNA Constructs for Vasa-mCherry-LOV, Vasa-mCherry, and Lifeact-GFP-ePDZ were previously used[44]. 1500 ng/μL stock of Vasa-mCherry-LOV/Vasa-mCherry was coinjected with 500 ng/μL stock of Lifeact-ePDZ and 750 ng/μL stock of TC-Vasa, 750 ng/μL stock of Nanos-GFP, or 300 ng/μL stock of β-catenin-GFP mRNAs. 8~16-cell embryos were irradiated with 12% 488 nm blue light on Olympus FV3000 or Nikon CSU-W1 Spinning disk confocal microscope for 0.5 s to recruit Vasa to the ectopic region during the 8~16-cell stage. Translation of TC-Vasa mRNA was detected by adding FlAsH -EDT$_2$ to the media right before the laser activation, followed by time-lapse imaging of every 1 min. Nanos-GFP or β-catenin-GFP was detected by time-lapse imaging at 15 min intervals up to the morula stage. Please also see Table 1 for a list of all resources used in this study.

**Data analysis.** In general, data are presented as mean ± s.e.m, and ****$P < 0.0001$, ***$P < 0.001$, **$P < 0.01$, *$P < 0.05$ throughout in this study. Statistical significance was analyzed by PRISM (GraphPad) using *t*-test or one-way ANOVA followed by Tukey post hoc test. Data calculations for specific experiments are described below.

**Vasa protein dynamics calculation.** Levels of Kaede-Vasa or Vasa-mCherry-LOV and FlAsH signals were measured on both sides of the mitotic spindle in animal and vegetal blastomeres during the 8~16-cell stage using Fluoview or NIS elements attached to the confocal laser microscopes or by *Image J*. Regions of interest of the same area on both sides of the spindle and used to measure average fluorescence intensity. This calculation was carried out for each time point and the resulting value divided by average fluorescence intensity immediately after photoconversion or optogenetic activation to determine the change in photoconverted Kaede-Vasa levels or in FlAsH signal over time. Excel was used to create a trendline for photoconverted Kaede-Vasa degradation during anaphase for both sides of the spindle. The slope for each trendline was determined through Excel, and the slope of opposite spindle sides was compared to each other. Each experiment was performed at least three independent times unless individually indicated.

**Immunofluorescence, OPP, and HPG calculations.** Eight- to 16-cell stage embryos were identified and levels of different proteins on the mitotic spindle in both the animal and vegetal blastomeres were measured at metaphase, anaphase, and telophase. Regions of interest of the same size were selected on both sides of the spindle, and the average intensity was measured through *ImageJ*. Three ROI's (cells) per embryo were quantified and averaged. The measurements for each stage were averaged and opposite sides of the spindle were compared to each other. The spindle/cytoplasm ratio within a cell, micromere/macromere ratio of averages in the vegetal blastomeres, and the side 1/side 2 ratio of averages in the animal blastomeres were graphed.

**TC tag detection, imaging, and calculations.** For TC experiments, embryos were injected with TC-Vasa-GFP or coinjected with membrane-mCherry, membrane-mCherry Vasa, or membrane-mCherry Vasa-ΔC-term, and TC-Vasa. At the 8~16-cell stage, embryos were transferred to a glass slide and the TC reagent (a biarsenical dye, a membrane-permeable and non-toxic reagent that is added to the media at any time point) was added for a final concentration of 3 μM. TC reagent exposed embryos were then imaged as sub-sections on the Zeiss LSM800 confocal microscope until the end of micromere formation at 5-sec intervals or as the whole-sections on the Olympus FV3000 confocal microscope at 1 min intervals.

For the pulse-chase experiment, embryos were injected with TC-Vasa mRNA and 3 μM ReAsH-EDT$_2$ added at the 8-cell stage for approximately 15, 30, or 45 min. After ReAsH -EDT$_2$ incubation, embryos were washed 3 times with 1X BAL buffer, and then FlAsH-EDT$_2$ was added to a final concentration of 3 μM. Embryos were then imaged at 5-s intervals for approximately 15 min.

For translational blocking by OPP treatment, embryos were injected with TC-GFP mRNA and OPP was added at the 8-cell stage to a final concentration of 20 μM. They were incubated in OPP for 30 min and then ReAsH -EDT$_2$ was added; embryos were imaged in 3 μM ReAsH -EDT$_2$ and 20 μM OPP. In this condition, translation was partially blocked, which allowed embryos to continue cell cycling during recording.

For signal calculations of TC-Vasa-GFP/Vasa-GFP injected embryos, the ReAsH signal on the micromere and macromere side of the spindle starting at metaphase of the 8~16-cell stage was measured by selecting an ROI of the same area on both sides of the spindle and using *Image J* to measure the ROI average fluorescence intensity at each time point. This calculation was carried out for each

time point until the end of the M-phase, then graphed and used to create a trend line on Excel. Graphs show the average trend lines of 2 (will be 3) embryos for each condition. For TC-Vasa + membrane-mCherry/-Vasa/-Vasa- ΔC-term injected embryos, Zen 2 imaging software was used to determine the red (mCherry) and green (TC-Vasa/FlAsH-EDT$_2$) fluorescence intensity profile at 6 points (3 through clumps/regions of high fluorescence intensity, 3 through regions of low fluorescence intensity) across the plasma membrane at time 2:55, 5:00, 6:20 min, 8:05 min, and 10:15 min for each condition. The fluorescence intensity of each wavelength across the membrane was measured for each of the red and green cortical signals. High intensity and low-intensity profiles were averaged on Excel and then graphed to illustrate the distance between red and green fluorescence peaks. For each injection condition, at least three embryos were analyzed in this manner. For embryos injected with TC-Vasa + Vasa-mCherry-LOV (or without LOV as a control) + Lifeact-ePDZ, the Fluoview software was used to analyze the signal level of FlAsH-EDT$_2$ at 1 min intervals on macromere- and micromere-side of the spindle.

**Statistics & reproducibility**. The sample size was determined by the condition that the analysis provides consistent trends across multiple experimental cycles. For technically challenging experiments, three representative embryos with the most consistent timing and angle across groups were analyzed in detail. To overcome the small sample size, multiple different experiments to address the same question was combined to make a single conclusion. Embryos showing significant defects were excluded from the analysis. For the detailed analysis with a small sample number, the embryos in a significantly different angle or timing were excluded from the analysis. Most of the experiments were repeated numerous times, yet only the embryos in the most consistent angle and timing across groups were chosen for analysis. All attempts at replication were successful. This article was contributed by multiple authors using the same or similar constructs and technologies multiple times across the article, which resulted in the same or similar results, providing natural randomization. Embryo handling and advanced live imaging require highly trained skills and eyes to confirm no technical mistake is involved in each experiment. Blinding was therefore not appropriate in this article.

**Reporting summary**. Further information on research design is available in the Nature Research Reporting Summary linked to this article.

## Data availability
This study does not include publicly available data or code. Any other datasets generated during and/or analyzed during the current study are available from the corresponding author on reasonable request. Source data for main figures are provided with this paper. Source data are provided with this paper.

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

## Acknowledgements

We thank Dr. Gary Wessel and the members of the PRIMO at Brown University for active discussions and critical reading of the manuscript. We would also like to thank Dr. Emily Cronin-Furman at Olympus Inc. for technical help in advanced imaging. This work was supported by the American Heart Association Scientist Development Grant (14SDG18350021), NIH (1R01GM126043-01), and NSF (IOS-1940975) to M.Y.

## Author contributions

A.F., J.P., A.U. were responsible for concepts, experimental design and undertaking, data analysis, and manuscript construction regarding experiments involving sea urchin embryos; M.Y. was responsible for concepts, experimental design and undertaking, data analysis, manuscript construction, and editing for all sections.

## Competing interests

The authors declare no competing interests.
