## [Peer Review File · Nature Communications]

Vasa nucleates asymmetric translation along the mitotic spindle during unequal cell divisionsReviewers' Comments:

Reviewer #1:

Remarks to the Author:

The manuscript by Fernandez-Nicolas et al presents a sound body of evidence supporting the conclusion that the DEAD-box helicase Vasa nucleates sites of localized translation in the early sea urchin embryo. Further, their data support a model whereby Vasa-mediated translation on the mitotic spindle in the vegetal cell divisions that produce micromeres produces spatial and temporal translational control that is essential for establishing asymmetry and specifying the micromeres at the progenitors of the germ line. The work in this paper represents a major advance over earlier work from the same investigators, which established that Vasa protein and vasa mRNA become enriched on mitotic spindles toward the future micromere domains, and that vasa knockdown causes a substantial reduction in general translation and failure of germ cell specification.

This work is highly significant both to developmental biology and to advancing our knowledge about spatiotemporal regulation of protein synthesis.

State-of-the-art optogenetic and imaging methods are used to analyze localized translation in real time and in living embryos, and the data are carefully quantitated throughout. This methodology more than meets the standards in the field.

For me, the big problem with the paper is the quality of presentation. Most of the figures are extremely busy, making it difficult to tease the key data that support the paper's conclusions out from a mass of controls, cartoons, and other information. For example, the key data in Figure 2 that shows asymmetric translation of Vasa is buried in panel B'. The use of primes and double primes to identify figure panels adds to the lack of clarity. Essentially all the figures need to be simplified so that the most important data are made readily apparent to the reader. This could be accomplished by removing some material (are all those time points really necessary in Fig 4E-G?) and moving a lot of the supporting information to supplemental material. The narrative in the text could also be tightened up and copyedited, as there are numerous examples of awkward English usage, misspellings, etc. Some of the figures also have misspellings in their included text.

More specific comments:

In Fig 1B' and 1C', what is the red signal in the top right of all the panels?

Also with respect to Fig 1, the loading control EMTB needs to be defined.

In Fig 3A', 3B', 3C' and all graphs in Fig S2 the values on the x-axis cannot be precisely aligned to the marks because of the inclusion of 'min' after each one. Just the numbers should be there. Also in 3B' and the right panel of S2A the scale on the x-axis does not appear to be linear.

In Fig 2C', 3B', 3C', S2A, S2C, S2D and perhaps others, equations, often complicated cubic and quadratic ones, are included in the graphs. How were these derived, at what confidence level do they describe the experimental data, and what relevance do they have to the conclusions of the paper? They are not discussed in the text.

On p. 5, line 25, the word 'translation' should replace 'protein'.

On p. 5, lines 28-29, how can there be both ~1.45 and 59% more protein synthesis?

On p. 6, line 4, the Lasko and Ashburner 1988 paper does not address phosphorylated RpS6. This should not be referenced here.

Referring to the C-terminal region of Vasa as the 'Dead region' is unfortunate, because it creates an obvious confusion with the DEAD-box helicase motifs that are found in a different part of the protein. I strongly recommend that the C-terminal region is simply referred to as such, or C-terminal domain if the authors prefer. Also Dehghani and Lasko 2015 (referenced in the text but reference missing) and 2016 demonstrated a functional role for the final C-terminal residues in *Drosophila* Vasa. This should be made more clear.

Finally, I found the cancer cell experiment to be very preliminary and not conducted with the same rigorous standard as the rest of the work in the paper. At a minimum the authors should address whether expressing membrane-bound DDX4 is toxic only to cancer cells that express endogenous DDX4, or if it is more generally toxic to normal cells or cells from cancers that do not express DDX4. Alternatively, Fig 7 and associated text could be dropped from the manuscript.

Reviewer #2:

Remarks to the Author:

In the manuscript entitled "Vasa nucleates asymmetric translation along the mitotic spindle during unequal cell division" the authors examine the role of protein translation during asymmetric cell division in the early sea urchin embryo. Asymmetric cell division (ACD) is a critical feature of development, where cell divisions may be geometrically asymmetric, developmentally asymmetric or both. Previous work from the corresponding author has focused on Vasa, a DEAD-box RNA helicase that regulates protein translation during development. In the sea urchin embryo, the fourth embryonic cell cleavage in the sea urchin is characterized by a quartet of vegetal blastomeres that divide asymmetrically to give rise to a set of smaller blastomeres (micromeres) that will give rise to skeletogenic mesoderm as well as the germ cells. Vasa associates with the mitotic spindle and becomes sequestered in the micromere lineage, and in this manuscript the authors use a variety of live cell imaging approaches to examine the mechanisms that regulate this enrichment. The authors demonstrate that spindle associated Vasa is not due to protein exchange with the cytoplasm, and that the increase in Vasa in smaller cell (micromere) during ACD was the result of spindle-associated translation. Indeed, spindle associated Vasa appears to be responsible for all new translation associated with the mitotic spindle. The authors perform a domain analysis to identify residues at the C-terminus that are essential for spindle targeting, and that mutants in this domain cannot rescue morpholino depletion of Vasa. Ectopic targeting of Vasa to the cortex using an optogenetic probe not only transferred the location of mitotic translation, but also disrupted developmental signaling, thus linking Vasa localization to the canonical Wnt signaling pathways that establish the developmental axes in this organism. Lastly, the authors report that this may be a conserved pathway, showing evidence for similar phenomena in human cultured cell lines.

This is a well-conceived and well-written study that tackles an outstanding issue in early development. Most work on Vasa has been performed in *Drosophila*, and this study takes advantage of a model system that has distinct advantages for studying ACD. Although the sea urchin work is quite convincing, the experiments in cultured cells described in Figure 7 and S8 were not convincing at all. Although it might have been simply image plane selection, but DDX4 did not appear to be particularly enriched on the spindle and the membrane-tagging experiments in Figure 7A were uninterpretable. The WT-DDX4 looked no different than the WT-membrane and the membrane-targeted, mutant DDX4 shows a cell post cytokinesis, where there is no central spindle and thus doesn't really show a cell in the same phase of cell division.

The authors need to better demonstrate that the Vasa/OPP signal is microtubule-associated for this data to be remotely believable. Lastly, although a minor point, the frenetic 3D rotations in the supplemental movies are distracting and don't help the viewer understand what is happening in 4D. This is particularly egregious in Supplemental movie 3.

Reviewer #3:

Remarks to the Author:

In the manuscript Fernandez-Nicolas et al. report on the role of the conserved helicase vasa in nucleating translational activity on the spindle on the micromere side when cells go from 8 cell to the 16-cell stage. They show that vasa-GFP is preferentially enriched at the micromere side of the spindle, and moreover that this spindle acts as a site of "major" translation in the cell. Using fluorescent probes for translation they show that the ectopic expression of vasa at the membrane leads to ectopic translation at the membrane. They also show that in some cancer cell lines, there is symmetric expression of the vasa homolog DDX4 in the spindle and forced expression of DDX4 at the membrane leads to translation at the membrane and cell mortality. There are previous studies that have shown asymmetric localization of mRNAs coding for regulatory factors (including translational regulators) in meiotic and mitotic spindles. The novelty of the current study is that the authors focus on vasa, a translational regulator, and moreover, they show that vasa can recruit a competent translational complex when ectopically expressed at the cell membrane. This is an interesting idea and experiments are challenging. But I have some concerns about the work that I outline below.

Using a vasa-GFP construct the authors show that during the formation of the micromere cells at the 16-cell stage, vasa-GFP accumulates in spindle inherited by the micromere cells. Using Kaede-vasa conversion the authors provide evidence that there is upregulated vasa-GFP at the vegetal spindle inherited by the micromeres. Using Kaede-vasa the authors also carry out experiments to show that while Kaede-vasa accumulates evenly across the spindles in animal pole blastomeres it is enriched in the spindle inherited by the micromeres. These data collected are from vasa-GFP, or Kaede-vasa and mCherry-EMTB mRNA injected embryos. In general, imaging in sea urchins produces crisp, high quality images. Overall, the resolution and quality of the images shown in this manuscript are not very good and in my opinion this will compromise the quality of the data collected for their analyses. Moreover, it is concerning that in several experiments the data shown is only from three embryos and in some cases the number of embryos examined is less than that. I understand that these are tough manipulations, but this is not a sufficient sample size for these experiments.

In figure 3, the authors report on experiments using Kaede-vasa to show that vasa does not translocate to the spindle during the M-phase, but does so at the beginning of M-phase entry. It is not clear what stage the embryos shown in Figure 3 A and B were imaged. These embryos look abnormal and do not appear to be at the 16-cell stage. It is not unusual to see sea urchin embryo undergoing early abnormal cleavages when microinjected with high concentrations of mRNA, and I am concerned that these embryos are not normal. It would help to have parallel bright field images of these embryos to assess if they are in fact normal or undergoing aberrant cell divisions. This brings up another concern about the medium that the authors are using to hold the embryos during imaging. On page 48 (bottom), the authors report that the still imaging of live embryos was done in 1X PBS. Imaging of sea urchin embryos is always done in normal seawater and culturing them in a low osmolality solution like 1X PBS would definitely lead to abnormal development and death.

In figure 4 the authors report on experiments where they used HPG or OPP to visualize general translational activities in the embryo. This is an interesting approach that to the best of knowledge has never been done in sea urchins before. But the images in this figure also are not compelling and in some cases, it is challenging to see the structures where the author's claim there is localized translation. For example, in Fig 4F I have difficulty seeing the spindle in the embryos shown.

In figure 5 the authors report on results of experiments where they use deletion constructs of vasa to identify the region of the protein required for localized translation on the spindle. Again, I am concerned about the poor quality of the images that makes this difficult to interpret. In fig 5E, the authors show membrane mCherry expression in a cell, but the fluorescence does not seem to have membrane localization. This odd localization is not explained in the manuscript.

In figure 6A and B, how many embryos were imaged in this experiment? The numbers are not reported. Also, what is the green fluorescence at the animal pole of the control embryo? This should be explained. In fig 6 D, D' embryos in D' appear to be at an earlier stage of development and looks abnormal in appearance. Also, what is the explanation for why is b-catenin becoming nuclearized in the animal pole blastomeres?

In figure 7 the authors report on the role of DDX4 in regulating translation in cancer cell lines. I think an important control here is to target vasa to membrane in cells that do not have vasa in mitotic spindle to show that this effect is not an artifact of vasa removing essential factors from the cytoplasm.

Other concerns:

In the methods section (pg 48, morpholino design) the authors mention injecting morpholinos into mouse zygotes. I assume they mean sea urchin zygotes?

Response to Reviewers' Comments

Dear Reviewers:

Thank you very much for your useful comments and suggestions. We took them to our heart and revised our manuscript accordingly. The major changes made in this version of the manuscript include removing human cancer data and transferring several data to Supplements to focus more on the main story. Individual changes made in the text are also highlighted in **Yellow**.

For your convenience, we enclosed a single PDF that contains all materials in one piece similar to the 1st submission. Further, we uploaded the individual text and figure files for this submission.

We hope this version of the manuscript reads better and meets the standard for publication in *Nature Communications*. Thank you again for your effort and support in this process.

Sincerely,
Mamiko Yajima, Ph.D.

Reviewer #1 (Remarks to the Author):

The manuscript by Fernandez-Nicolas et al presents a sound body of evidence supporting the conclusion that the DEAD-box helicase Vasa nucleates sites of localized translation in the early sea urchin embryo. Further, their data support a model whereby Vasa-mediated translation on the mitotic spindle in the vegetal cell divisions that produce micromeres produces spatial and temporal translational control that is essential for establishing asymmetry and specifying the micromeres at the progenitors of the germline. The work in this paper represents a major advance over earlier work from the same investigators, which established that Vasa protein and vasa mRNA become enriched on mitotic spindles toward the future micromere domains and that vasa knockdown causes a substantial reduction in general translation and failure of germ cell specification.

This work is highly significant both to developmental biology and to advancing our knowledge about spatiotemporal regulation of protein synthesis.

State-of-the-art optogenetic and imaging methods are used to analyze localized translation in real-time and in living embryos, and the data are carefully quantitated throughout. This methodology more than meets the standards in the field.

For me, the big problem with the paper is the quality of the presentation. Most of the figures are extremely busy, making it difficult to tease the key data that support the paper's conclusions out from a mass of controls, cartoons, and other information. For example, the key data in Figure 2 that shows asymmetric translation of Vasa is buried in panel B". The use of primes and double primes to identify figure panels adds to the lack of clarity. Essentially all the figures need to be simplified so that the most important data are made readily apparent to the reader. This could be accomplished by removing some material (are all those time points really necessary in Fig 4E-G?) and moving a lot of the supporting information to supplemental material. The narrative in the text could also be tightened up and copyedited, as there are numerous examples of awkward English usage, misspellings, etc. Some of the figures also have misspellings in their included text.

- Thank you for the promising comments. We moved many more figures to the supplements to simplify the main story in this version of the manuscript. The manuscript was actually copy-edited by the professional yet we went through the text more carefully this time.

More specific comments:

In Fig 1B' and 1C', what is the red signal in the top right of all the panels?

- The red signal is from the software to show the direction. Since this appears not helpful in this manuscript, we removed them in this revised manuscript.

Also with respect to Fig 1, the loading control EMTB needs to be defined.

- Thank you. We clarified it in the figure legend.

In Fig 3A", 3B", 3C" and all graphs in Fig S2 the values on the x-axis cannot be precisely aligned to the marks because of the inclusion of 'min' after each one. Just the numbers should be there. Also in 3B" and the right panel of S2A the scale on the x-axis does not appear to be linear.

- Thank you for the suggestion. We revised the graphs and corrected errors as suggested.

In Fig 2C", 3B", 3C", S2A, S2C, S2D and perhaps others, equations, often complicated cubic and quadratic ones, are included in the graphs. How were these derived, at what confidence level do they describe the experimental data, and what relevance do they have to the conclusions of the paper? They are not discussed in the text.

- Illustration and equation were meant to help explain the analysis. We explained more details in the figure legend yet if this is rather confusing, we will be happy to remove those equations.

On p. 5, line 25, the word 'translation' should replace 'protein'.

- Corrected as suggested

On p. 5, lines 28-29, how can there be both ~1.45 and 59% more protein synthesis?

- We re-visited our calculations and corrected them. Thank you for pointing it out.

On p. 6, line 4, the Lasko and Ashburner 1988 paper does not address phosphorylated RpS6. This should not be referenced here.

Referring to the C-terminal region of Vasa as the 'Dead region' is unfortunate, because it creates an obvious confusion with the DEAD-box helicase motifs that are found in a different part of the protein. I strongly recommend that the C-terminal region is simply referred to as such, or C-terminal domain if the authors prefer. Also, Dehghani and Lasko 2015 (referenced in the text but reference missing) and 2016 demonstrated a functional role for the final C-terminal residues in *Drosophila* Vasa. This should be made more clear.

- We replaced Vasa-Dead with Vasa- Δ C-term as suggested.

- Our apologies - "Dehghani and Lasko 2015" was a typo of "Dehghani and Lasko 2016". We corrected this error, thank you for pointing this out. We also revised to highlight the above report more specifically as suggested. Of note, however, this reference was not yet published when we started this study, thus we performed thorough deletion experiments without having any biased view in this study. We then later noticed the same C-terminal region is important both in *Drosophila* and sea urchin, which gave us further excitement and confidence in this finding.

Finally, I found the cancer cell experiment to be very preliminary and not conducted with the same rigorous standard as the rest of the work in the paper. At a minimum the authors should address whether expressing membrane-bound DDX4 is toxic only to cancer cells that express endogenous DDX4, or if it is more generally toxic to normal cells or cells from cancers that do not express DDX4. Alternatively, Fig 7 and associated text could be dropped from the manuscript.

- We agreed and removed any experiments related to cancer cells from this manuscript to simplify the story.

Reviewer #2 (Remarks to the Author):

In the manuscript entitled "Vasa nucleates asymmetric translation along the mitotic spindle during unequal cell division" the authors examine the role of protein translation during asymmetric cell division in the early sea urchin embryo. Asymmetric cell division (ACD) is a critical feature of the development, where cell divisions may be geometrically asymmetric, developmentally asymmetric, or both. Previous work from the corresponding author has focused on Vasa, a DEAD-box RNA helicase that regulates protein translation during development. In the sea urchin embryo, the fourth embryonic cell cleavage in the sea urchin is characterized by a quartet of vegetal blastomeres that divide asymmetrically to give rise to a set of smaller blastomeres (micromeres) that will give rise to skeletogenic mesoderm as well as the germ cells. Vasa associates with the mitotic spindle and becomes sequestered in the micromere lineage, and in this manuscript, the authors use a variety of live cell imaging approaches to examine the mechanisms that regulate this enrichment. The authors demonstrate that spindle-associated Vasa is not due to protein exchange with the cytoplasm and that the increase in Vasa in smaller cells (micromere) during

ACD was the result of spindle-associated translation. Indeed, spindle-associated Vasa appears to be responsible for all new translation associated with the mitotic spindle. The authors perform a domain analysis to identify residues at the C-terminus that are essential for spindle targeting, and that mutants in this domain cannot rescue morpholino depletion of Vasa. Ectopic targeting of Vasa to the cortex using an optogenetic probe not only transferred the location of mitotic translation but also disrupted developmental signaling, thus linking Vasa localization to the canonical Wnt signaling pathways that establish the developmental axes in this organism. Lastly, the authors report that this may be a conserved pathway, showing evidence for similar phenomena in human cultured cell lines.

This is a well-conceived and well-written study that tackles an outstanding issue in early development. Most work on Vasa has been performed in *Drosophila*, and this study takes advantage of a model system that has distinct advantages for studying ACD. An while the sea urchin work is quite convincing, the experiments in cultured cells described in Figure 7 and S8 were not convincing at all. Although it might have been simply image plane selection, but DDX4 did not appear to be particularly enriched on the spindle and the membrane-tagging experiments in Figure 7A were uninterpretable. The WT-DDX4 looked no different than the WT-membrane and the membrane-targeted, mutant DDX4 shows a cell post cytokinesis, where there is no central spindle and thus doesn't really show a cell in the same phase of cell division. The authors need to better demonstrate that the Vasa/OPP signal is microtubule-associated for this data to be remotely believable. Lastly, although a minor point, the frenetic 3D rotations in the supplemental movies are distracting and don't help the viewer understand what is happening in 4D. This is particularly egregious in Supplemental movie 3.

- Thank you for the encouraging comments. We decided to remove cancer cell data in Figure 7 from this manuscript as suggested by Reviewer 1. We have another full manuscript underway that focuses only on human DDX4 that addresses the questions the Reviewer raised more comprehensively. Therefore, we decided to integrate this part of the data into another upcoming paper to simplify the story in this revised manuscript.

- 3D rotation is used to confirm the depth of imaging and/or area of photoconversion but not necessarily to track the dynamics as 2D projection is superior for the latter purpose as the reviewer suggests. We understand this is only a technical point and may not be of every reader's interest. We emphasized this point in the Supplementary Movie Legends, so only the interested readers may watch the movies.

Reviewer #3 (Remarks to the Author):

In the manuscript Fernandez-Nicolas et al. report on the role of the conserved helicase vasa in nucleating translational activity on the spindle on the micromere side when cells go from 8 cell to the 16-cell stage. They show that vasa-GFP is preferentially enriched at the micromere side of the spindle, and moreover that this spindle acts as a site of "major" translation in the cell. Using fluorescent probes for translation they show that the ectopic expression of vasa at the membrane leads to ectopic translation at the membrane. They also show that in some cancer cell lines, there is symmetric expression of the vasa homolog DDX4 in the spindle and forced expression of DDX4 at the membrane leads to translation at the membrane and cell mortality. There are previous studies that have shown asymmetric localization of mRNAs coding for regulatory factors (including translational regulators) in meiotic and mitotic spindles. The novelty of the current study is that the authors focus on vasa, a translational regulator, and moreover, they show that vasa can recruit a competent translational complex when ectopically expressed at the cell membrane. This is an interesting idea and experiments are challenging. But I have some concerns about the work that I outline below.

Using a vasa-GFP construct the authors show that during the formation of the micromere cells at the 16-cell stage, vasa-GFP accumulates in spindle inherited by the micromere cells. Using Kaede-vasa conversion the authors provide evidence that there is upregulated vasa-GFP at the vegetal spindle inherited by the micromeres. Using Kaede-vasa the authors also carry out experiments to show that while Kaede-vasa accumulates evenly across the spindles in animal pole blastomeres it is enriched in the spindle inherited by the micromeres. These data collected are from vasa-GFP, or Kaede-vasa and mCherry-EMTB mRNA injected embryos. In general, imaging in sea urchins produces crisp, high quality images. Overall, the resolution and quality of the images shown in this manuscript are not very good and in my opinion this will compromise the quality of the data collected for their analyses. Moreover, it is concerning that in several experiments the data shown is only from three embryos and in some cases the number of embryos examined is less than that. I understand that these are tough manipulations, but this is not a sufficient sample size for these experiments.

- Each data has at least 3 embryos and 2 independent cycles of experiments. As for real-time imaging, many of these images were taken for 50-90 slices per minute for double colors, which is needed for faithful quantitative analysis in

developing embryos. If we scanned only a single or few z-slices by using higher laser power and long exposure time, the signal would have looked much clearer. However, the purpose of these experiments is to perform quantitative analysis for the fluorescent signals of interest without bleaching the signals, damaging the cells, or missing the timing. To achieve this goal, we explored a quite few microscopes and technologies and identified the fast scanning approach (e.g. Resonant) gives the best outcomes for this purpose as it provides sufficient speed, minimum bleaching, and fine (but not best) resolution to track the signals in rapidly developing embryos. Therefore, we are challenging the technology and would greatly appreciate the reviewer's understanding in this matter. Similarly, each file produces a huge amount of data and substantial time is needed to construct and analyze each data. We of course tried more than 3 embryos or 3 cycles of experiments to confirm the trend, yet selected the most consistent angle and timing of the embryo across sample groups to be compared for formal analysis. Further, to support our conclusions, we rather focused on various different approaches not limited to Kaede-Vasa experiments as shown in the rest of the Figures to have better confidence in our conclusions. We also included these Kaede-Vasa results with the hope to showcase that quantitative real-time imaging is technically possible in the rapidly developing embryo with the current technology. We believe that this approach will be useful not just for this study but for any future studies that focus on the subcellular dynamics of the proteins of interest in the embryo.

In figure 3, the authors report on experiments using Kaede-vasa to show that vasa does not translocate to the spindle during the M-phase, but does so at the beginning of M-phase entry. It is not clear what stage the embryos shown in Figure 3 A and B were imaged. These embryos look abnormal and do not appear to be at the 16-cell stage. It is not unusual to see sea urchin embryo undergoing early abnormal cleavages when microinjected with high concentrations of mRNA, and I am concerned that these embryos are not normal. It would help to have parallel bright field images of these embryos to assess if they are in fact normal or undergoing aberrant cell divisions. This brings up another concern about the medium that the authors are using to hold the embryos during imaging.

On page 48 (bottom), the authors report that the still imaging of live embryos was done in 1X PBS. Imaging of sea urchin embryos is always done in normal seawater and culturing them in a low osmolality solution like 1X PBS would definitely lead to abnormal development and death.

- As the reviewer suggests embryos are highly sensitive and could fail in cell cycling due to various reasons not limited to the microinjection. Therefore, we only selected the blastomeres with normal cell cycling. If they were abnormal, they would have failed in normal cell cycling, which would also prevent us from analyzing the data. For Fig. 3 (now Fig. S3), we also selected the cells facing toward the objective as they provide the best signal, therefore the angle of the embryo was not considered in this experiment. Embryos were also lightly squashed by the cover glass to completely immobilize during 4D imaging. Therefore some embryos may look unfamiliar due to the angle. Lastly, as mentioned above, including extra time for brightfield is not possible even with the current high-speed scanner. As the technology advances, we will be able to image more colors within a limited time frame and with better resolution, but this is the current technological limit and we would appreciate the reviewer's understanding in this regard.

-Page 48, Still imaging in PBS was performed for fixed embryos. We corrected this error. Thank you for pointing this out.

In figure 4 the authors report on experiments where they used HPG or OPP to visualize general translational activities in the embryo. This is an interesting approach that to the best of knowledge has never been done in sea urchins before. But the images in this figure also are not compelling and in some cases, it is challenging to see the structures where the author's claim there is localized translation. For example, in Fig 4F I have difficulty seeing the spindle in the embryos shown.

- Fig .4A-B (now Fig. 4a & d) are OPP and HPG, and Fig 4F (now Fig. 4h) is the TC-Vasa result. All experiments use Vasa as a counter-staining as we are here analyzing the OPP or HPG signal associated with Vasa on the spindle. We emphasized this point in the text. We also provided the OPP image counterstained with tubulin to compare in Fig. S4c. We also hope the un-compressed images in this revision help reduce the concerns.

In figure 5 the authors report on results of experiments where they use deletion constructs of vasa to identify the region of the protein required for localized translation on the spindle. Again, I am concerned about the poor quality of the images that makes this difficult to interpret. In fig 5E, the authors show membrane mCherry expression in a cell, but the fluorescence does not seem to have membrane localization. This odd localization is not explained in the manuscript.

- As mentioned above, all images were taken with a z-stack. Depending on the plane and the thickness of the stack the appearance may change. Vasa makes relatively large granules, which make the image look less smooth under the high contrast condition.

In figure 6A and B, how many embryos were imaged in this experiment? The numbers are not reported. Also, what is the green fluorescence at the animal pole of the control embryo? This should be explained. In fig 6 D, D' embryos in D' appear to be at an earlier stage of development and looks abnormal in appearance. Also, what is the explanation for why is b-catenin becoming nuclearized in the animal pole blastomeres?

- We added the number of embryos observed in the figure legends. Thank you for pointing this out.

- The bright green fluorescence is a junk that typically shows non-specific fluorescence by microscopy. We added this explanation in the figure legend.

- From our view, it is around the same developmental stage yet the angle is slightly different. Further, as mentioned, the embryo is lightly squashed during imaging, which can cause unfamiliar looking depending on the angle. Due to so many technical hurdles to cross, in these experiments, we selected the embryos that successfully formed micromeres for the analyses. We added these explanations in the figure legend.

In figure 7 the authors report on the role of DDx4 in regulating translation in cancer cell lines. I think an important control here is to target vasa to membrane in cells that do not have vasa in mitotic spindle to show that this effect is not an artifact of vasa removing essential factors from the cytoplasm.

- We removed this experiment from this manuscript as suggested by Reviewer1. We are indeed testing Vasa's differential function in different somatic cells in another manuscript, therefore this data will fit better in another full manuscript.

Other concerns:

In the methods section (pg 48, morpholino design) the authors mention injecting morpholinos into mouse zygotes. I assume they mean sea urchin zygotes?

- Thank you for pointing it out. We corrected this error.

Reviewers' Comments:

Reviewer #1:

Remarks to the Author:

My previous comments have been mostly addressed and I think the presentation is much improved. Only a few minor points remain:

The legend to Fig 3c refers to $R^{>2}$ but the graphs show equations that do not include $R^{>2}$. This needs to be corrected.

In Figs S3 and S4, 'EMBRYO' is misspelled throughout, and the complicated equations that I recommended for removal last time remain in place. I still recommend removal of these.

'Uninjected' is misspelled in Fig S9.

'enrichment' is misspelled in Fig S10.

Reviewer #2:

Remarks to the Author:

In the revised manuscript entitled "Vasa nucleates asymmetric translation along the mitotic spindle during unequal cell division" the authors examine the role of protein translation during asymmetric cell division in the early sea urchin embryo. In preparing the revision, the authors considered the comments of all three reviewers, and considerably revised the manuscript such that the figures are simplified and the mammalian tissue culture work has been cut from the manuscript. Aside from a few clarifications, that text did not require significant revisions, only the organization of the data. The manuscript has been improved by the authors' changes and is now ready for publication.

Reviewer #3:

Remarks to the Author:

This is a much improved version of the manuscript and the authors have done well to improve the figures. I am still concerned about the low number of embryos used in some experiments. But I agree with the authors that they are pushing the boundaries of the technology, and that their results are significant. I am satisfied with the revisions.

REVIEWERS' COMMENTS

Reviewer #1 (Remarks to the Author):

My previous comments have been mostly addressed and I think the presentation is much improved. Only a few minor points remain:

The legend to Fig 3c refers to R^2 but the graphs show equations that do not include R^2 . This needs to be corrected.

In Figs S3 and S4, 'EMBRYO' is misspelled throughout, and the complicated equations that I recommended for removal last time remain in place. I still recommend removal of these.

- We removed equations from figures and R^2 from the legend.

'Uninjected' is misspelled in Fig S9.

'enrichment' is misspelled in Fig S10.

- Thank you very much for pointing out errors. We collected all typos and errors.

Reviewer #2 (Remarks to the Author):

In the revised manuscript entitled "Vasa nucleates asymmetric translation along the mitotic spindle during unequal cell division" the authors examine the role of protein translation during asymmetric cell division in the early sea urchin embryo. In preparing the revision, the authors considered the comments of all three reviewers, and considerably revised the manuscript such that the figures are simplified and the mammalian tissue culture work has been cut from the manuscript. Aside from a few clarifications, that text did not require significant revisions, only the organization of the data. The manuscript has been improved by the authors' changes and is now ready for publication.

- Thank you for the kind comments.

Reviewer #3 (Remarks to the Author):

This is a much improved version of the manuscript and the authors have done well to improve the figures. I am still concerned about the low number of embryos used in some experiments. But I agree with the authors that they are pushing the boundaries of the technology, and that their results are significant. I am satisfied with the revisions.

- We are glad to hear the current data are satisfactory.